# PI-DUAL: USING PRIVILEGED INFORMATION TO DISTINGUISH CLEAN FROM NOISY LABELS

## ABSTRACT

Label noise is a pervasive problem in deep learning that often compromises the generalization performance of trained models. Recently, leveraging privileged information (PI) – information available only during training but not at test time – has emerged as an effective approach to mitigate this issue. Yet, existing PI-based methods have failed to consistently outperform their no-PI counterparts in terms of preventing overfitting to label noise. To address this deficiency, we introduce Pi-DUAL, an architecture designed to harness PI to distinguish clean from wrong labels. Pi-DUAL decomposes the output logits into a prediction term, based on conventional input features, and a noise-fitting term influenced solely by PI. A gating mechanism steered by PI adaptively shifts focus between these terms, allowing the model to implicitly separate the learning paths of clean and wrong labels. Empirically, Pi-DUAL achieves significant performance improvements on key PI benchmarks (e.g., $+6.8\%$ on ImageNet-PI), establishing a new state-of-the-art test set accuracy. Additionally, Pi-DUAL is a potent method for identifying noisy samples post-training, outperforming other strong methods at this task. Overall, Pi-DUAL is a simple, scalable and practical approach for mitigating the effects of label noise in a variety of real-world scenarios with PI.

## 1 INTRODUCTION

Many deep learning models are trained on large noisy datasets, as obtaining cleanly labeled datasets at scale can be expensive and time consuming (Snow et al., 2008; Sheng et al., 2008). However, the presence of label noise in the training set tends to damage generalization performance as it forces the model to learn spurious associations between the input features and the noisy labels (Zhang et al., 2017; Arpit et al., 2017). To mitigate the negative effects of label noise, recent methods have primarily tried to prevent overfitting to the noisy labels, often utilising the observation that neural networks tend to first learn the clean labels before memorizing the wrong ones (Maennel et al., 2020; Baldock et al., 2021). For instance, these methods include filtering out incorrect labels, correcting them, or enforcing regularization on the training dynamics (Han et al., 2018; Liu et al., 2020; Li et al., 2020b). Other works, instead, try to capture the noise structure in an input-dependent fashion (Patrini et al., 2017; Liu et al., 2022; Collier et al., 2022; 2023).

The above methods are however designed for a standard supervised learning setting, where models are tasked to learn an association between input features $\boldsymbol{x} \in \mathbb{R}^d$ and targets $y \in \{1, \ldots, K\}$ (assuming $K$ classes) from a training set of pairs $\{(\boldsymbol{x}_i, \tilde{y}_i)\}_{i \in [n]}$ of features and (possibly) noisy labels $\tilde{y} \in \{1, \ldots K\}$. As a result, they need to model the noise in the targets as a function of $\boldsymbol{x}$. Yet, in many practical situations, the mistakes introduced during the annotation process may not solely depend on the input $\boldsymbol{x}$, but rather be mostly explained by annotation-specific side information, such as the experience of the annotator or the attention they paid while annotating. For this reason, a recent line of work (Vapnik & Vashist, 2009; Collier et al., 2022; Ortiz-Jimenez et al., 2023) has proposed to use *privileged information* (PI) to mitigate the affects of label noise. PI is defined as additional features available at training time but not at test time. It can include annotation features such as the annotator ID, the amount of time needed to provide the label, or their experience.

Remarkably, having access to PI at training time, even when it is not available at test time, has been shown to be an effective tool for dealing with instance-dependent label noise. Most notably, Ortiz-Jimenez et al. (2023) showed that by exploiting PI it is possible to activate positive learning

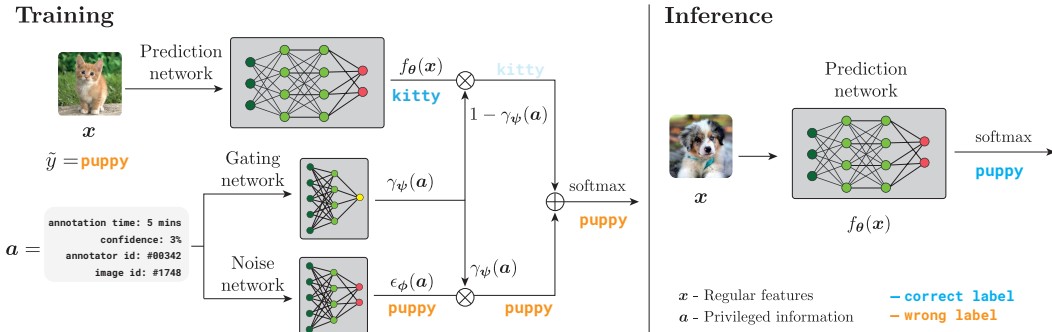

Figure 1: **Illustration of the architecture of Pi-DUAL.** (Left) During training, Pi-DUAL fits the noisy target label $\tilde{y}$ combining the output of a prediction network (which takes the regular features $\boldsymbol{x}$ as input) and a noise network (which takes PI $\boldsymbol{a}$ as input). The outputs of these sub-networks are weighted based on the output of a gating network (which also has $\boldsymbol{a}$ as input) and then passed through a softmax operator to obtain the predictions. (Right) During inference, when only $\boldsymbol{x}$ is available, Pi-DUAL does not need access to PI and simply uses the prediction network to predict the clean target $y$.

shortcuts to memorize, and therefore explain away, noisy training samples, thereby improving generalization. Nevertheless, and perhaps surprisingly, current PI-based methods do not systematically outperform no-PI baselines in the presence of label noise, making them a less competitive alternative in certain cases (Ortiz-Jimenez et al., 2023).

In this work, we aim to improve the performance of PI strategies by proposing a new PI-guided noisy label architecture: **Pi-DUAL**, a **P**rivileged **I**nformation network to **D**istinguish **U**ntrustworthy **A**nnotations and **L**abels. Specifically, during training, we propose to decompose the output logits into a weighted combination of a prediction term, that depends only on the regular features $\boldsymbol{x}$, and a noise-fitting term, that depends only on the PI features $\boldsymbol{a} \in \mathbb{R}^p$. Pi-DUAL toggles between these terms using a gating mechanism, also solely a function of $\boldsymbol{a}$, that decides if a sample should be learned primarily by the prediction network, or explained away by the noise network (see Fig. 1). This dual sub-network design adaptively routes the clean and wrong labels through the prediction and noise networks so that they are fit based on $\boldsymbol{x}$ or $\boldsymbol{a}$, respectively. This protects the prediction network from overfitting to the label noise. Pi-DUAL is simple to implement, effective, and can be trained end-to-end with minimal modifications to existing training pipelines. Unlike some previous methods Pi-DUAL also scales to training on very large datasets. Finally, in public benchmarks for learning with label noise, Pi-DUAL achieves state-of-the-art results on datasets with rich PI features (+4.5% on CIFAR-10H, +1.3% ImageNet-PI (low-noise) and +6.8% on ImageNet-PI (high-noise)); and performs on par with previous methods on benchmarks with weak PI or no PI at all, despite not being specifically designed to work in these regimes.

Overall, the main contributions of our work are:

- We present Pi-DUAL, a novel PI method to combat label noise based on a dual path architecture that implicitly separates the noisy fitting path from the clean prediction path during training.

- We show that Pi-DUAL achieves strong performance on noisy label benchmarks, and outperforms previous state-of-the-art methods when given access to high-quality PI features.

- Through extensive ablation studies, we show the benefits of Pi-DUAL in preventing the memorization of wrong labels by the test-time prediction network.

- We show that Pi-DUAL can also serve as a strong method to detect wrong labels.

In summary, our work advances the state-of-the-art in noisy label learning by effectively leveraging privileged information through the novel Pi-DUAL architecture. Pi-DUAL can be easily integrated into any learning pipeline, requires minimal hyperparameters, and can be trained end-to-end in a single stage. Overall, Pi-DUAL is a scalable and practical approach for mitigating the effects of label noise in a variety of real-world scenarios with PI.

## 2 RELATED WORK

Noisy label methods mostly fall into two broad categories: those that explicitly model the noise signal, and those that rely on implicit network dynamics to correct or ignore the wrong labels (Song et al., 2022). Noise modeling techniques aim to learn the function that governs the noisy annotation process explicitly during training, inverting it during inference to obtain the clean labels. Some methods model the annotation function using a transition matrix (Patrini et al., 2017); others model uncertainty via a heteroscedastic noise term (Collier et al., 2021; 2023); and recently, some works explicitly parameterize the label error signal as a vector for each sample in the training set (Tanaka et al., 2018; Yi & Wu, 2019; Liu et al., 2022). Implicit-dynamics based approaches, on the other hand, operate under the assumption that wrong labels are harder to learn than the correct labels (Zhang et al., 2017; Maennel et al., 2020). Using this intuition, different methods have come up with different heuristics to correct (Jiang et al., 2018; Han et al., 2018; Yu et al., 2019) or downweight (Liu et al., 2020; Menon et al., 2020; Bai et al., 2021) the influence of wrong labels during training. This has sometimes led to very complex methods that require multiple stages of training (Patrini et al., 2017; Bai et al., 2021; Albert et al., 2023; Wang et al., 2023), higher computational cost (Han et al., 2018; Jiang et al., 2018; Han et al., 2018; Yu et al., 2019), and many additional parameters that do not scale well to large datasets (Yi & Wu, 2019; Liu et al., 2020; 2022).

The introduction of privileged information (PI) offers an alternative dimension to tackle the noisy label problem (Hernández-Lobato et al., 2014; Lopez-Paz et al., 2015; Collier et al., 2022). In this regard, Ortiz-Jimenez et al. (2023) showed that most PI methods work as implicit-dynamics approaches. They rely on the use of PI to enable learning shortcuts, to avoid memorizing the incorrect labels using the regular features. Moreover, these approaches are attractive for their scalability, as they usually avoid the introduction of extra training stages or parameters. However, current PI methods can sometimes lag behind in performance with respect to no-PI baselines. The main reason is that these methods still try to learn the noise predictive distribution $p(\tilde{y}|\boldsymbol{x})$ by marginalizing $\boldsymbol{a}$ in $p(\tilde{y}|\boldsymbol{x}, \boldsymbol{a})$, when they should actually aim to learn the clean distribution $p(y|\boldsymbol{x})$ directly. However, prior PI methods do not have an explicit mechanism to identify or correct the wrong labels.

Our proposed method, Pi-DUAL, tries to circumvent these issues by explicitly modeling the *clean distribution*, exploiting the ability of PI to distinguish clean and wrong labels. Our design allows Pi-DUAL to scale effectively across large datasets and diverse class distributions, while maintaining high performance and low training complexity as seen in Tab. 1. We further note that our design is reminiscent of mixtures of experts (MoE) that were shown to be a competitive architecture for language modeling (Shazeer et al., 2017) and computer vision (Riquelme et al., 2021). By analogy, we can see Pi-DUAL as an MoE containing a single MoE layer with two heterogeneous experts—the prediction and noise networks—located at the logits of the model and with a dense gating.

Table 1: Comparison of different representative methods to learn with label noise *vs* Pi-DUAL on several design axis: ability to leverage PI, ability to explicitly model the noise signal, parameter scalability, and whether training requires multiple models and training stages.

| Methods | Leverage PI | Explicit noise modeling | Scalability (#samples) | Scalability (#classes) | Training complexity |
|---|---|---|---|---|---|
| Forward-T (Patrini et al., 2017) | ✗ | ✓ | ✓ | ✗ | 1 model, 2 stages |
| Co-Teaching (Han et al., 2018) | ✗ | ✗ | ✓ | ✓ | 2 models, 1 stage |
| Divide-Mix (Li et al., 2020b) | ✗ | ✗ | ✓ | ✓ | 2 models, 1 stage |
| ELR (Liu et al., 2020) | ✗ | ✗ | ✗ | ✗ | 1 model, 1 stage |
| SOP (Liu et al., 2022) | ✗ | ✓ | ✗ | ✗ | 1 model, 1 stage |
| HET-XL (Collier et al., 2023) | ✗ | ✓ | ✓ | ✓ | 1 model, 1 stage |
| Distill. PI (Lopez-Paz et al., 2015) | ✓ | ✗ | ✓ | ✓ | 2 model, 2 stage |
| AFM (Collier et al., 2022) | ✓ | ✗ | ✓ | ✓ | 1 model, 1 stage |
| TRAM++ (Ortiz-Jimenez et al., 2023) | ✓ | ✗ | ✓ | ✓ | 1 model, 1 stage |
| Pi-DUAL (Ours) | ✓ | ✓ | ✓ | ✓ | 1 model, 1 stage |

## 3 PI-DUAL

### 3.1 NOISE MODELING

In traditional supervised learning, we typically assume that there exists a groundtruth function $f^\star : \mathcal{X} \to \mathcal{Y}$ which maps input features $\boldsymbol{x} \in \mathcal{X}$ to labels $y \in \mathcal{Y}$ where $\mathcal{X} = \mathbb{R}^d$ and $\mathcal{Y} = \{1, \ldots, K\}$. However, the labels in real-world scenarios are usually gathered via a noisy annotation process.

In this work, we model this annotation process as a function of some, possibly unknown, side information $\boldsymbol{a} \in \mathcal{A}$, which explains away the noise from the training labels. This side information could be anything, from the experience of the annotator, to their intrinsic motivation. The important modeling aspect is that given this side information one should be able to tell whether a label is incorrect or not, and the type of mistake that was made. We can model this process mathematically as a function $h : \mathcal{X} \times \mathcal{A} \to \mathcal{Y}$ that maps the input features and the side information to the noisy human label $\tilde{y}$. We assume that the mistakes in the annotation process depend only on $\boldsymbol{a}$, i.e.,

$$\tilde{y} = h(\boldsymbol{x}, \boldsymbol{a}) = [1 - \gamma(\boldsymbol{a})]f^\star(\boldsymbol{x}) + \gamma(\boldsymbol{a})\epsilon(\boldsymbol{a}) \tag{1}$$

Here $\gamma : \mathcal{A} \to \{0, 1\}$ acts as a switch between clean and wrong labels, and $\epsilon : \mathcal{A} \to \mathcal{Y}$ models the incorrect labelling function. Consequently, the training dataset $\mathcal{D}$ consists of two types of training samples $\mathcal{D}_{\text{correct}} = \{(\boldsymbol{x}, \tilde{y}) \in \mathcal{D} \mid \tilde{y} = f^\star(\boldsymbol{x})\}$ and $\mathcal{D}_{\text{wrong}} = \{(\boldsymbol{x}, \tilde{y}) \in \mathcal{D} \mid \tilde{y} = \epsilon(\boldsymbol{a})\}$. In this regard, when training a network to map $\boldsymbol{x}$ to $\tilde{y}$ on $\mathcal{D} = \mathcal{D}_{\text{correct}} \cup \mathcal{D}_{\text{wrong}}$, we are effectively asking it to learn to different target functions, where only one of them depends on $\boldsymbol{x}$. This forces the network to memorize part of the training data, which therefore hurts its generalization (Zhang et al., 2017).

In practice, however, we will not have access to the exact side information, and we will be able to rely at most on meta-data, and PI, about the annotation process. That is, we consider a learning problem in which our training data consists of triplets $(\boldsymbol{x}, \tilde{y}; \boldsymbol{a})$ where $\boldsymbol{a} \in \mathbb{R}^p$ is a vector of PI features such as high latency features related to the annotation process e.g., annotator experience, or even a random vector introduced to model unobserved features (Ortiz-Jimenez et al., 2023). We present here our method that uses this setting to explicitly model $h$ and learn effectively in the presence of large amounts of label noise in the training set.

### 3.2 METHOD DESCRIPTION

Based on the noise model from before, we propose Pi-DUAL, a novel PI-based architecture designed to mimic the generative noise model proposed in Eq. (1). Specifically, during training, Pi-DUAL factorizes its ouptut logits into two terms, i.e.,

$$h_{\boldsymbol{\theta}, \boldsymbol{\phi}, \boldsymbol{\psi}}(\boldsymbol{x}, \boldsymbol{a}) = [1 - \gamma_{\boldsymbol{\psi}}(\boldsymbol{a})]f_{\boldsymbol{\theta}}(\boldsymbol{x}) + \gamma_{\boldsymbol{\psi}}(\boldsymbol{a})\epsilon_{\boldsymbol{\phi}}(\boldsymbol{a}) \tag{2}$$

where $f_{\boldsymbol{\theta}} : \mathcal{X} \to \mathbb{R}^c$ represents a *prediction network* tasked with approximating the ground truth labelling function $f^\star$ and $\epsilon_{\boldsymbol{\phi}} : \mathcal{A} \to \mathbb{R}^c$ a *noise network*, modeling the noise signal $\epsilon$. Here, $\gamma_{\boldsymbol{\phi}}$ denotes a *gating network* tasked with learning the discriminating mechanism $\gamma$, where we apply a sigmoid activation function to the output to restrict $\gamma_{\boldsymbol{\psi}}(\boldsymbol{a})$ to be within in $[0, 1]$. Moreover, following the recommendations of Ortiz-Jimenez et al. (2023), we augment the available PI features with a unique random identifier for each training sample to help the network explain away the missing factors of the noise using this identifier. The dimension of this vector, known as random PI length, is the only additional hyperparameter we tune for Pi-DUAL. During inference, when PI is not available, Pi-DUAL relies solely on $f_{\boldsymbol{\theta}}(\boldsymbol{x})$ to predict the clean label $y$ (see Fig. 1).

The dual gated logit structure of Pi-DUAL is reminiscent of sparsely-gated mixture of experts which also factorize its predictions at the logit level, albeit providing the same input $\boldsymbol{x}$ to each expert (Shazeer et al., 2017). Pi-DUAL instead provides $\boldsymbol{x}$ and $\boldsymbol{a}$ to different networks, which effectively decouples learning the task-specific samples and the noise-specific samples with different features. Indeed, assuming that the incorrect labels are independent of $\boldsymbol{x}$ and that the noise is only a function of the PI $\boldsymbol{a}$, there will always be a natural tendency by the network to use $\epsilon_{\boldsymbol{\phi}}(\boldsymbol{a})$ to explain away those labels that it cannot easily learn with $f_{\boldsymbol{\theta}}(\boldsymbol{x})$. The gating network $\gamma_{\boldsymbol{\phi}}$ facilitates this separation by utilizing the discriminative power of the PI to guide this process. In Sec. 5.3, we ablate all these elements of the architecture to show that they all contribute to learning the clean labels.

Pi-DUAL has multiple advantages over previous PI methods like TRAM or AFM (Collier et al., 2022). Indeed, previous methods tend to directly expose the no-PI term $f_{\boldsymbol{\theta}}(\boldsymbol{x})$ to the noisy labels,

e.g., through $\mathcal{L}(f_{\boldsymbol{\theta}}(\boldsymbol{x}), \tilde{y})$, which can thus lead to an overfitting to the noisy labels based on $\boldsymbol{x}$. In contrast, Pi-DUAL instead solves

$$\underset{\boldsymbol{\theta}, \boldsymbol{\phi}, \boldsymbol{\psi}}{\text{minimize}} \sum_{(\boldsymbol{x}, \tilde{y}; \boldsymbol{a}) \in \mathcal{D}} \mathcal{L}\left(\text{softmax}\left(h_{\boldsymbol{\theta}, \boldsymbol{\phi}, \boldsymbol{\psi}}(\boldsymbol{x}, \boldsymbol{a})\right), \tilde{y}\right), \quad (3)$$

and never explicitly forces $f_{\boldsymbol{\theta}}(\boldsymbol{x})$ to fit all $\tilde{y}$'s. This allows the model to predict the clean label for all the training samples without incurring a loss penalty, as it can fit the residual noise signal using $\epsilon_{\boldsymbol{\phi}}(\boldsymbol{a})$. In Sec. 5.1 we analyze in detail these dynamics.

Another important advantage of Pi-DUAL is that it explicitly learns to model the noise signal in the training set. This makes it more interpretable than implicit-dynamics methods like TRAM, and puts it on par with state-of-the-art noise-modeling methods. However, because Pi-DUAL can leverage PI to model the noise signal, it exhibits a much better detection performance than no-PI methods, while at the same time allowing it to scale to datasets with millions of datapoints, as it does not require to store individual parameters for each sample in the training set to effectively learn the label noise. In Sec. 4.3, we compare Pi-DUAL to other state-of-the-art methods to detect wrong labels.

### 3.3 THEORETICAL INSIGHTS

To further support the design of Pi-DUAL described in Eq. (2), we study the theoretical behavior of the predictor $h_{\boldsymbol{\theta}, \boldsymbol{\phi}, \boldsymbol{\psi}}(\boldsymbol{x}, \boldsymbol{a})$ within a simplified linear regression setting. More specifically, we consider the setting where the clean and noisy targets are respectively generated from two Gaussian distributions $\mathcal{N}(\boldsymbol{x}^\top \boldsymbol{w}^\star, \sigma^2)$ and $\mathcal{N}(\boldsymbol{a}^\top \boldsymbol{v}^\star, \sigma^2)$, for two weight vectors $(\boldsymbol{w}^\star, \boldsymbol{v}^\star)$ parameterizing linearly their means. In this tractable setting, we show that Pi-DUAL is a robust estimator in the presence of label noise as its risk depends less severely on the number of wrong labels. We summarize below the main insights of our analysis, and details are available in Appendix A.

We compare two estimators, Pi-DUAL and an ordinary least squares estimator (OLS) that ignores the side information $\boldsymbol{a}$. We show that in terms of their abilities to generalize on *clean targets*—as measured by their risks (Bach, 2021)—Pi-DUAL exhibits a more robust behavior. In particular, while the risk of OLS tends to be proportional to the number of wrong labels $|\mathcal{D}_{\text{wrong}}|$, Pi-DUAL has a risk that more gracefully scales with respect to the number of examples that the gates $\gamma_{\boldsymbol{\psi}}$ fail to identify. Our experiments in Sec. 5.2 show that, in practice, the gates learned by Pi-DUAL typically manage to identify the clean and wrong labels.

## 4 EXPERIMENTAL RESULTS

We now validate the effectiveness of Pi-DUAL on several public noisy label benchmarks with PI and compare it extensively to other algorithms. We show that Pi-DUAL achieves (a) state-of-the-art results on clean test accuracy and noise detection tasks (especially when there is good PI available) and (b) scales up to datasets with millions of examples.

### 4.1 EXPERIMENTAL SETTINGS

Our experimental settings follow the benchmarking practices laid out by Ortiz-Jimenez et al. (2023). In particular, we use the same architectures, training schedules and public codebase (Nado et al., 2021) to perform all our experiments. In terms of baseline choices, in order to achieve a fair comparison, we compare Pi-DUAL to our own implementations of the methods in Tab. 1 that use only one model and one stage of training[1]. Moreover, to further ensure fairness, we use on each dataset the same architecture and the same training strategy across all compared methods. For each result, we perform a grid search over hyperparameters. Notably, while other methods require tuning at least two additional hyperparameters on top of the cross-entropy baseline; Pi-DUAL only requires tuning the random PI length, making its tuning budget smaller. We use a *noisy validation set*, held-out from the training set, to select the best hyperparameters and report results over the clean test set. We provide more details of the experimental setting in Appendix B.

Pi-DUAL does not require to use early stopping to achieve strong results as it does not suffer from overfitting issues (see Fig. 2). However, early stopping is essential to achieve good performance for

---

[1]We do not run ELR and SOP on ImageNet-PI as they require 1 billion extra parameters (see Appendix B.2).

Table 2: Test accuracy of different methods on noisy label datasets with PI. We report mean and standard deviation accuracy over multiple runs with the best hyperparameters and early-stopping.

|  | Methods | CIFAR-10H (worst) | CIFAR-10N (worst) | CIFAR-100N (fine) | ImageNet-PI (low-noise) | ImageNet-PI (high-noise) |
|---|---|---|---|---|---|---|
| No-PI | Cross-entropy | $51.1_{\pm2.2}$ | $80.6_{\pm0.2}$ | $60.4_{\pm0.5}$ | $68.2_{\pm0.2}$ | $47.2_{\pm0.2}$ |
|  | ELR | $48.5_{\pm1.4}$ | $\mathbf{86.6}_{\pm0.7}$ | $\mathbf{64.0}_{\pm0.3}$ | - | - |
|  | HET | $50.8_{\pm1.4}$ | $81.9_{\pm0.4}$ | $60.8_{\pm0.4}$ | $69.4_{\pm0.1}$ | $51.9_{\pm0.0}$ |
|  | SOP | $51.3_{\pm1.9}$ | $85.0_{\pm0.8}$ | $61.9_{\pm0.6}$ | - | - |
| PI | TRAM | $64.9_{\pm0.8}$ | $80.5_{\pm0.5}$ | $59.7_{\pm0.3}$ | $69.4_{\pm0.2}$ | $54.0_{\pm0.1}$ |
|  | TRAM++ | $66.8_{\pm0.3}$ | $83.9_{\pm0.2}$ | $61.1_{\pm0.2}$ | $69.5_{\pm0.0}$ | $53.8_{\pm0.3}$ |
|  | AFM | $64.0_{\pm0.6}$ | $82.0_{\pm0.3}$ | $60.0_{\pm0.2}$ | $70.3_{\pm0.0}$ | $55.3_{\pm0.2}$ |
| Pi-DUAL (Ours) | | $\mathbf{71.3}_{\pm3.3}$ | $84.9_{\pm0.4}$ | $\mathbf{64.2}_{\pm0.3}$ | $\mathbf{71.6}_{\pm0.1}$ | $\mathbf{62.1}_{\pm0.1}$ |

the other methods. We, thus, always report results at the epoch with the best accuracy on the noisy validation set. In Appendix C.4, we provide results for all methods without early stopping.

Our experiments are conducted on five noisy datasets with realistic label noise, either derived from a noisy human annotation process or produced by imperfect model predictions. A summary of the main features of each datasets is shown in Appendix B.1. Importantly, we note that CIFAR-10H (Peterson et al., 2019), ImageNet-PI (low-noise) and ImageNet-PI (high-noise) all have excellent-quality PI features at the sample level that seem to capture important information of the annotation process. On the other hand, CIFAR-10N and CIFAR-100N (Wei et al., 2022) provide low-quality PI, in the form of averages over batches of samples, which does not have enough resolution to distinguish clean and wrong labels at the sample level (Ortiz-Jimenez et al., 2023). Despite this, Pi-DUAL still performs comparatively to no-PI methods on those datasets.

### 4.2 PREDICTING THE CLEAN LABELS

Tab. 2 reports the test accuracy of Pi-DUAL compared to previous noisy label methods, averaged over 5 and 3 random seeds for CIFAR and ImageNet-PI, respectively. As we can see, Pi-DUAL achieves state-of-the-art performance on the three datasets with high quality PI. It improves by $4.5\%$ over the most competitive PI baseline on CIFAR-10H and by 20 points over the best performing no-PI methods. It also achieves a 1.3 point and 6.8 point lead on ImageNet-PI low-noise and high-noise, respectively. These are remarkable results given the 1000 classes in ImageNet-PI and the scale of these datasets. Indeed, they show that Pi-DUAL can effectively leverage PI in these settings to distinguish between correct and wrong labels during training, while learning the clean labels with the prediction network.

On the other hand, on the two datasets with low quality PI, we observe that Pi-DUAL achieves better results than previous PI methods by more than 3 points on CIFAR-100N. It also performs comparatively with no-PI methods, even though the quality of the PI does not allow to properly distinguish between clean and wrong labels (see Sec. 5.2).

### 4.3 DETECTION OF WRONG LABELS

We validate the ability of Pi-DUAL to detect the wrong labels in the training set, allowing practitioners to relabel those instances, or filter them out in future runs. Our detection method is very simple: It consists in obtaining confidence estimates of the prediction network on the noisy label $\tilde{y}$ set, i.e., $\mathrm{softmax}(f_\theta(\boldsymbol{x}))[\tilde{y}]$, for the training samples and thresholding them to distinguish the correct and incorrect ones. Indeed, because $f_\theta$ of Pi-DUAL only learns to confidently predict clean labels $y$ during training, its confidence on the noisy label is a very good proxy for a noise indicator: If the confidence is high, then it is likely that $\tilde{y} = y$; but if it is low, then probably $\tilde{y}$ is wrong.

Tab. 3 shows the area under the receiver operating characteristic curve (AUC) obtained by applying this simple detection method with different methods on all our PI benchmarks. As we can see, Pi-DUAL achieves the best results by a large margin in all datasets except CIFAR-10N (where it performs comparatively to the best methods). These performance gains are a clear sign that Pi-

DUAL can effectively minimize the amount of overfitting of the prediction network to the noisy labels. In most cases, the prediction network has a very low confidence (near 0%) on the observed noisy labels, while having a very high confidence (near 100%) on the samples with clean labels. In Appendix C.3, we show the distribution of prediction confidences on all datasets.

In our experiments, we observe that the simple confidence thresholding is a strong detection method across all datasets. However, we also evaluated the ability of the gating network $\gamma_\psi$ in detecting the wrong labels. As we see in Tab. 3, thresholding the gate's outputs is also an effective method for noise detection, which can even outperform confidence thresholding on certain datasets, i.e., CIFAR-10H. However, we observe that the performance of gate thresholding suffers more than confidence thresholding on datasets with low-quality PI (e.g. CIFAR-10N). As we will see in Sec. 5.2, this is due to the fact that, in those datasets, the gating network cannot exploit the PI to discriminate easily between correct and wrong labels. Still, this does not prevent the prediction network from learning the clean distribution, and thus its detection ability does not suffer as much. Choosing which of the two methods to use is, in general, a dataset-dependent decision: If there is good PI available gate thresholding achieves the best results, but confidence thresholding performs well overall, so we recommend it as a default choice.

Table 3: AUC of different noise detection methods based on confidence thresholding of the network predictions on noisy labels or thresholding of the gating network's output (for Pi-DUAL).

| Methods | CIFAR-10H (worst) | CIFAR-10N (worst) | CIFAR-100N (fine) | ImageNet-PI (low-noise) | ImageNet-PI (high-noise) |
|---|---|---|---|---|---|
| Cross-entropy | 0.810 | 0.951 | 0.883 | 0.935 | 0.941 |
| ELR | 0.745 | **0.968** | 0.876 | - | - |
| SOP | 0.808 | 0.964 | 0.889 | - | - |
| TRAM++ | 0.834 | 0.955 | 0.883 | 0.937 | 0.959 |
| Pi-DUAL (conf.) | 0.954 | 0.962 | **0.911** | **0.953** | **0.986** |
| Pi-DUAL (gate) | **0.982** | 0.808 | 0.726 | 0.952 | **0.986** |

## 5 FURTHER ANALYSIS

In this section, we provide further analysis on the training dynamics of Pi-DUAL, the distribution of the learned gates and several ablations on our method. Overall, we show that Pi-DUAL behaves as expected from its design, and that all pieces of its architecture contribute to its good performance.

### 5.1 TRAINING DYNAMICS

To verify that Pi-DUAL effectively decouples the learning paths of samples with correct and wrong labels, we study the training dynamics of the prediction and noise networks on each of these sets of samples, in comparison to the training dynamics of cross-entropy baseline. We observe in Fig. 2 that the prediction network of Pi-DUAL mostly fits the correct labels, as its training accuracy on samples with wrong labels is always very low on all datasets[2]. Meanwhile, the noise network shows the opposite behavior and mostly fits the wrong labels on CIFAR-10H and CIFAR-100N. Interestingly, we observe that the noise network does not fit any samples on ImageNet-PI. We attribute this behavior to the fact that ImageNet has more than a million samples and 1000 classes, so fitting the noise is very hard. Indeed, as shown on the bottom row of Fig. 2, the cross-entropy baseline also ignores the samples with wrong labels. However, the cross-entropy baseline has lower training accuracy on the correct labels than Pi-DUAL as it cannot effectively separate the two distributions, and therefore achieves worse test accuracy.

In all datasets, we see that the test accuracy of Pi-DUAL grows gradually and steadily with training and that overfitting to the wrong labels does not hurt its performance as these are mostly fit by the noise network $\epsilon_\phi$. Meanwhile, we observe that on CIFAR-10H and CIFAR-100N, the test accuracy of the cross-entropy baseline starts degrading as the accuracy on samples with wrong labels starts to grow. This is a clear sign that Pi-DUAL effectively leverages the PI to learn shortcuts that protect the feature extraction of $f_\theta$ and therefore does not require to use early-stopping to achieve its best results.

---

[2]Results for other datasets are shown in Appendix C.1.

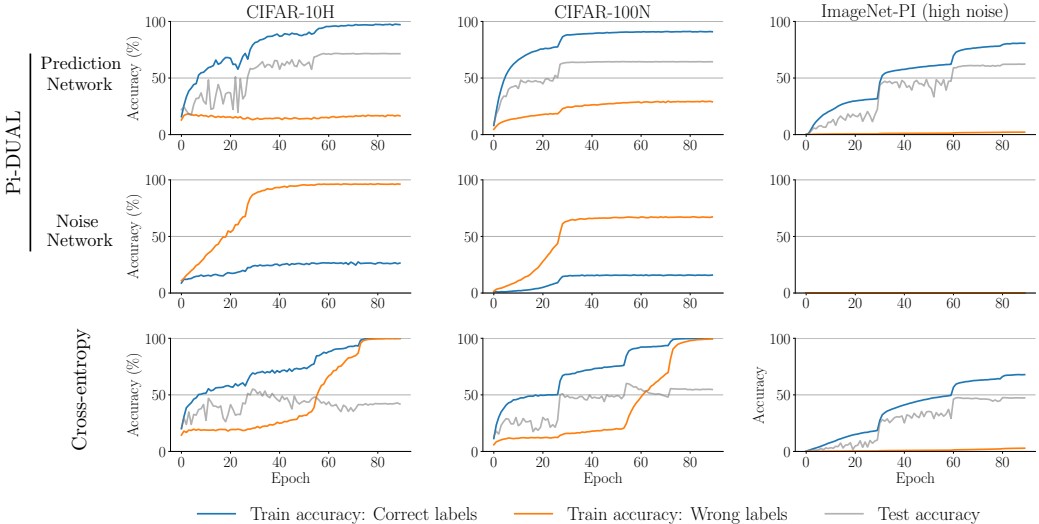

Figure 2: Training curves of Pi-DUAL and the cross-entropy baseline on different datasets. The first two rows shows the training dynamics of the prediction network and noise network, respectively. We plot separately the training accuracy on clean and wrong labels and the test accuracy.

## 5.2 ANALYSIS OF THE GATING NETWORK PREDICTIONS

In our model, the gating network $\gamma_\psi$ is tasked with learning the binary indicator signal $\gamma$, which tells whether a sample belongs to $\mathcal{D}_{\text{correct}}$ or $\mathcal{D}_{\text{wrong}}$. To show that the model works as intended, we plot in Fig. 3 the distribution of $\gamma_\psi(a)$ separately for samples with correct and wrong labels after training on different datasets[3]. As expected, in the two datasets with high-quality PI – CIFAR-10H and ImageNet-PI – the gate distribution achieves a remarkable degree of separation between the two distributions (cf. Tab. 3). And even in the case of CIFAR-100N, where the PI is not very informative, the gate output still separates a big portion of the wrong labels.

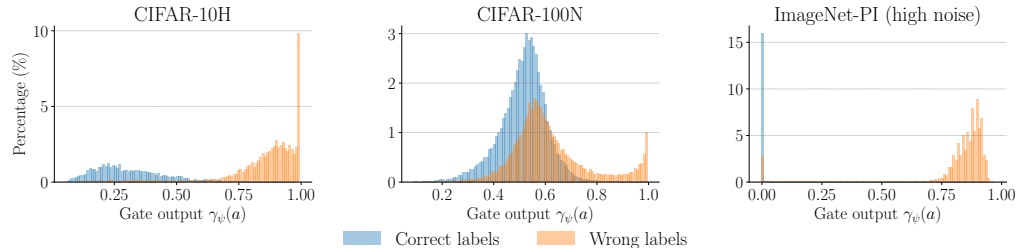

Figure 3: Value of $\gamma_\psi(a)$ over training samples with correct and wrong labels on several datasets.

To give a better intuition of what Pi-DUAL learns, we provide some visual examples of both success and failures cases of the gating network when training on ImageNet-PI (high-noise). As shown in Fig. 4, the gating network can often detect blatantly wrong annotations which are further corrected by the prediction network. Interestingly, we observe that in the few cases where the gating network makes a mistake, the predicted clean label is not so far from what many humans would suggest – like in the crane picture on the bottom right which is recognized by Pi-DUAL as a fire truck.

## 5.3 ABLATION STUDIES

We finally present the results from various ablation studies analysing the contributions of the different components of the Pi-DUAL design in Tab. 4.

---

[3]Results for other datasets are shown in Appendix C.2.

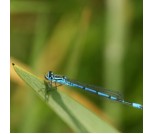
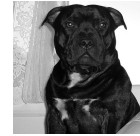
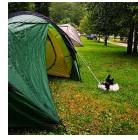
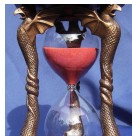
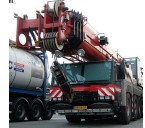

Success examples

label: picket fence
pred: planetarium
$\gamma_\psi(a)$ : 0.97

label: oboe
pred: damselfly
$\gamma_\psi(a)$ : 0.97

label: cinema
pred: Staffordshire bullterrier
$\gamma_\psi(a)$ : 0.97

Failure examples

label: mountain tent
pred: border collie
$\gamma_\psi(a)$ : 0.95

label: hourglass
pred: thimble
$\gamma_\psi(a)$ : 0.94

label: crane
pred: fire truck
$\gamma_\psi(a)$ : 0.93

Figure 4: Examples of ImageNet-PI images that the gating network suggests are mislabeled. The first row shows samples with actually wrong labels, and the second row shows examples with correct labels but assumed to be wrong by the gating network. Here, "label" denotes the annotation label $\tilde{y}$ and "pred" the prediction by $f_\theta$.

Table 4: Test accuracy of various ablation studies over Pi-DUAL on the different PI datasets.

| Ablations | CIFAR-10H (worst) | CIFAR-10N (worst) | CIFAR-100N (fine) | ImageNet-PI (low-noise) | ImageNet-PI (high-noise) |
|---|---|---|---|---|---|
| Cross-entropy | $51.1_{\pm 2.2}$ | $80.6_{\pm 0.2}$ | $60.4_{\pm 0.5}$ | $68.2_{\pm 0.2}$ | $47.2_{\pm 0.2}$ |
| Pi-DUAL | $\mathbf{71.3}_{\pm 3.3}$ | $\mathbf{84.9}_{\pm 0.4}$ | $\mathbf{64.2}_{\pm 0.3}$ | $\mathbf{71.6}_{\pm 0.1}$ | $\mathbf{62.1}_{\pm 0.1}$ |
| (no gating network) | $61.5_{\pm 1.2}$ | $\mathbf{84.5}_{\pm 0.2}$ | $59.0_{\pm 0.2}$ | $67.9_{\pm 0.1}$ | $47.8_{\pm 0.8}$ |
| (no noise network) | $59.7_{\pm 3.6}$ | $82.4_{\pm 1.0}$ | $59.7_{\pm 0.3}$ | $\mathbf{71.6}_{\pm 0.2}$ | $\mathbf{62.3}_{\pm 0.1}$ |
| (gate in prob. space) | $62.2_{\pm 1.3}$ | $81.6_{\pm 0.8}$ | $59.4_{\pm 1.1}$ | $71.0_{\pm 0.1}$ | $60.4_{\pm 0.1}$ |
| (only random PI) | $53.5_{\pm 2.2}$ | $83.7_{\pm 1.3}$ | $61.8_{\pm 0.3}$ | $68.4_{\pm 0.1}$ | $47.0_{\pm 0.4}$ |

**Architecture ablation.** Pi-DUAL gives $a$ as input to two networks: the noise network $\epsilon_\phi$ and the gating network $\gamma_\psi$. As shown in Tab. 4, removing either of the two elements from the architecture generally results in lower performance gains than with the full architecture. Interestingly, on ImageNet-PI, the noise network does not seem to be critical. We attribute this behavior to the fact that, on these datasets, Pi-DUAL does not need to overfit to the noisy labels to achieve good performance (cf. Fig. 2). Indeed, just using the gating mechanism to toggle on-off the fitting of the noisy labels seems sufficient to achieve good performance in a dataset with so many classes.

**Gating in probability space.** In Sec. 3.2 we chose to parameterize Pi-DUAL in the logit space. Another alternative, would have been to parameterize the gating mechanism in the probability space. We see, however, that although the probabilistic version of Pi-DUAL performs better than the cross-entropy baseline in most cases, it underperforms compared to the logit space version.

**Performance without PI.** We argued before that Pi-DUAL performs better on datasets with high-quality PI as this permits to wield the most power from its structure. For completeness, we now test the performance of Pi-DUAL without access to dataset-specific PI features. That is, having only access to the random PI sample-identifier proposed by Ortiz-Jimenez et al. (2023). We see that without access to PI, Pi-DUAL still can perform better than the cross-entropy baseline, but its performance deteriorates significantly, i.e., having access to good PI is fundamental for Pi-DUAL's success.

## 6 CONCLUSION

In this paper, we have presented Pi-DUAL, a new method that utilizes PI to combat label noise by introducing a dual network structure designed to model the generative process of the noisy annotations. Experimental results have demonstrated the effectiveness of Pi-DUAL in learning to fit the clean label distribution, and to detect noisy samples. Pi-DUAL sets a new state-of-the-art in datasets with high-quality PI. We have performed extensive ablation studies and thorough empirical and theoretical analysis to give insights into how Pi-DUAL works. Importantly, Pi-DUAL is very easy to implement and can be plugged into any training pipeline. It can also be scaled up to datasets with millions of examples and thousands of classes. Moving forward it will be interesting to study extensions of Pi-DUAL that can also tackle other problems with PI beyond supervised classification.

## ETHICS STATEMENT

Overall, we do not see any special ethical concerns stemming directly from our work. In particular, we note that although annotator IDs are part of the PI used in our experiments, none of our results require the use of personally identifiable annotator IDs. In fact, cryptographically safe IDs in the form of hashes work equally well as PI. In this regard, we do not think that there are serious concerns about possible identity leakages stemming from our work if the proper anonymization protocols are followed.

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
