## A  THEORETICAL INSIGHTS: RISK ANALYSIS

**Model and notations.**  We assume the following regression setting

$$\boldsymbol{y} = \begin{bmatrix} \boldsymbol{y}_1 \\ \boldsymbol{y}_2 \end{bmatrix} = \begin{bmatrix} \boldsymbol{X}_1 \boldsymbol{w}^\star \\ \boldsymbol{0} \end{bmatrix} + \begin{bmatrix} \boldsymbol{0} \\ \boldsymbol{A}_2 \boldsymbol{v}^\star \end{bmatrix} + \begin{bmatrix} \varepsilon_1 \\ \varepsilon_2 \end{bmatrix} \in \mathbb{R}^n$$

where we have $n = n_1 + n_2$ observations such that

- $\boldsymbol{y}_1 = \boldsymbol{X}_1 \boldsymbol{w}^\star + \varepsilon_1 \in \mathbb{R}^{n_1}$ with $\boldsymbol{X}_1 \in \mathbb{R}^{n_1 \times d}$ and $\varepsilon_1 \sim \mathcal{N}(\boldsymbol{0}, \sigma^2 \boldsymbol{I})$,
- $\boldsymbol{y}_2 = \boldsymbol{A}_2 \boldsymbol{v}^\star + \varepsilon_2 \in \mathbb{R}^{n_2}$ with $\boldsymbol{A}_2 \in \mathbb{R}^{n_2 \times m}$ and $\varepsilon_2 \sim \mathcal{N}(\boldsymbol{0}, \sigma^2 \boldsymbol{I})$.

The vector $\boldsymbol{y}_1$ corresponds to the clean targets that depend on the features $\boldsymbol{X}_1$ while $\boldsymbol{y}_2$ corresponds to the noisy targets that are explained by the privileged information (PI) represented by $\boldsymbol{A}_2$.

We use the matrix forms $\boldsymbol{X} = [\boldsymbol{X}_1, \boldsymbol{X}_2] \in \mathbb{R}^{n \times d}$, $\boldsymbol{A} = [\boldsymbol{A}_1, \boldsymbol{A}_2] \in \mathbb{R}^{n \times m}$ and $\varepsilon = [\varepsilon_1, \varepsilon_2] \in \mathbb{R}^n$. Moreover, we consider the diagonal mask matrix $\boldsymbol{\gamma}^\star \in \{0, 1\}^{n \times n}$ such that

$$\boldsymbol{\gamma}^* \boldsymbol{X} = \begin{bmatrix} \boldsymbol{X}_1 \\ \boldsymbol{0} \end{bmatrix} \quad \text{and} \quad (\boldsymbol{I} - \boldsymbol{\gamma}^*) \boldsymbol{A} = \begin{bmatrix} \boldsymbol{0} \\ \boldsymbol{A}_2 \end{bmatrix}.$$

We list below some notation that we will repeatedly use

- The covariance matrices $\boldsymbol{Q} = \boldsymbol{X}^\top \boldsymbol{X}$ and $\boldsymbol{Q}_1 = \boldsymbol{X}_1^\top \boldsymbol{X}_1$
- The difference between the contributions of the standard features and the PI features

$$\boldsymbol{\delta}^\star = \boldsymbol{X} \boldsymbol{w}^\star - \boldsymbol{A} \boldsymbol{v}^\star \in \mathbb{R}^n$$

- The orthogonal projector onto the span of the columns of $\boldsymbol{X}$:

$$\boldsymbol{\Pi}_x = \boldsymbol{X} (\boldsymbol{X}^\top \boldsymbol{X})^{-1} \boldsymbol{X}^\top \in \mathbb{R}^{n \times n}.$$

- For any diagonal mask matrix $\boldsymbol{\gamma} \in \{0, 1\}^{n \times n}$, we define the diagonal matrix that records the differences with respect to the reference $\boldsymbol{\gamma}^\star$

$$\boldsymbol{\Delta}_\gamma = \boldsymbol{\gamma}^\star - \boldsymbol{\gamma} \in \{-1, 0, 1\}^{n \times n}.$$

The rest of our exposition follows the structure of Collier et al. (2022).

### A.1  DEFINITION OF THE RISK

To compare different predictors, we will consider their *risks*, that is, their ability to generalize. We focus on the *fixed* design analysis (Bach, 2021), i.e., we study the errors only due to resampling the additive noise $\varepsilon$. In our context, we are more specifically interested in *the performance of the predictors on the clean targets* (with predictors having been trained on both clean and noisy targets).

Formally, given a predictor $\boldsymbol{\theta}$ based on the training quantities $(\boldsymbol{X}, \boldsymbol{A}, \varepsilon)$, we consider

$$\boldsymbol{y}_1' = \boldsymbol{X}_1 \boldsymbol{w}^\star + \varepsilon_1'$$

where the prime $'$ is to show the difference with the training quantities without prime, and we define the risk of $\boldsymbol{\theta}$ as

$$\mathcal{R}(\boldsymbol{\theta}) = \mathbb{E}_{\varepsilon_1' \sim p(\varepsilon_1')} \left\{ \frac{1}{n_1} \| \boldsymbol{y}_1' - \boldsymbol{X}_1 \boldsymbol{\theta} \|^2 \right\}. \tag{4}$$

A simple expansion of the square with $\mathbb{E}_{\varepsilon'}[\|\varepsilon_1'\|^2] = n_1 \sigma^2$ leads to the standard expression

$$\mathcal{R}(\boldsymbol{\theta}) = \frac{1}{n_1} \| \boldsymbol{X}_1 (\boldsymbol{\theta} - \boldsymbol{w}^\star) \|^2 + \sigma^2 = \frac{1}{n_1} \| \boldsymbol{\gamma}^\star \boldsymbol{X} (\boldsymbol{\theta} - \boldsymbol{w}^\star) \|^2 + \sigma^2. \tag{5}$$

To obtain the final expression of the risk, we eventually take a second expectation $\mathbb{E}_{\varepsilon \sim p(\varepsilon)}[\mathcal{R}(\boldsymbol{\theta})]$ with respect to the training quantity $\varepsilon$ (Bach, 2021).

## A.2 Main result

We state below our main result and discuss its implications.

**Proposition A.1** *Consider some diagonal mask matrix $\boldsymbol{\gamma} \in \{0, 1\}^{n \times n}$ and the masked versions of $\boldsymbol{X}$ and $\boldsymbol{A}$ which we refer to as $\bar{\boldsymbol{X}} = \boldsymbol{\gamma}\boldsymbol{X}$ and $\bar{\boldsymbol{A}} = (\boldsymbol{I} - \boldsymbol{\gamma})\boldsymbol{A}$.*

*Let us assume that $\boldsymbol{Q} \in \mathbb{R}^{d \times d}$, $\bar{\boldsymbol{X}}^\top \bar{\boldsymbol{X}} \in \mathbb{R}^{d \times d}$, $\bar{\boldsymbol{A}}^\top \bar{\boldsymbol{A}} \in \mathbb{R}^{m \times m}$ and*

$$\begin{bmatrix} \bar{\boldsymbol{X}}^\top \bar{\boldsymbol{X}} & \bar{\boldsymbol{X}}^\top \bar{\boldsymbol{A}} \\ \bar{\boldsymbol{A}}^\top \bar{\boldsymbol{X}} & \bar{\boldsymbol{A}}^\top \bar{\boldsymbol{A}} \end{bmatrix} \in \mathbb{R}^{(d+m) \times (d+m)} \tag{6}$$

*are all invertible. Let us define by $\boldsymbol{w}_0$ the ordinary-least-squares predictor (see Eq. (8)). Similarly, let us define by $\boldsymbol{w}_1$ the Pi-DUAL predictor, using $\boldsymbol{\gamma}$ as (pre-defined) gates (see Eq. (10)).*

*It holds that the risk $\mathbb{E}[\mathcal{R}(\boldsymbol{w}_0)]$ of $\boldsymbol{w}_0$ is larger than the risk $\mathbb{E}[\mathcal{R}(\boldsymbol{w}_1)]$ of $\boldsymbol{w}_1$ if and only if*

$$\|\boldsymbol{\gamma}^\star \boldsymbol{\Pi}_x (\boldsymbol{I} - \boldsymbol{\gamma}^*)\boldsymbol{\delta}^\star\|^2 + \sigma^2 \mathrm{tr}(\boldsymbol{Q}^{-1}\boldsymbol{Q}_1) > \|\boldsymbol{\gamma}^\star \boldsymbol{X} \bar{\boldsymbol{H}} \boldsymbol{\Delta}_\gamma \boldsymbol{\delta}^\star\|^2 + \sigma^2 \mathrm{tr}(\bar{\boldsymbol{Q}}_a^{-1}\boldsymbol{Q}_1) \tag{7}$$

*where the matrices $\bar{\boldsymbol{H}}$ and $\bar{\boldsymbol{Q}}_a$ are defined in Section A.4.*

The proofs of the risk expressions can be found in Sections A.3 and A.4.

### A.2.1 Discussion

The condition in Eq. (7) brings into play the bias terms and the variance terms of the risks of $\boldsymbol{w}_0$ and $\boldsymbol{w}_1$.

As intuitively expected, the variance term corresponding to $\boldsymbol{w}_0$ is smaller than that of $\boldsymbol{w}_1$. Indeed, Pi-DUAL requires to learn more parameters (both $\boldsymbol{w}$ and $\boldsymbol{v}$) than in the case of the standard ordinary least squares. More precisely, if the spans of the columns $\bar{\boldsymbol{A}}$ and $\bar{\boldsymbol{X}}$ are close to be orthogonal to each other (as suggested by the invertibility condition for Eq. (6)), we approximately have

$$\mathrm{tr}(\boldsymbol{Q}^{-1}\boldsymbol{Q}_1) \approx d\frac{n_1}{n} < \mathrm{tr}(\bar{\boldsymbol{Q}}_a^{-1}\boldsymbol{Q}_1) \approx d\frac{n_1}{\bar{n}_1}$$

where $\bar{n}_1 = \boldsymbol{1}^\top \boldsymbol{\gamma} \boldsymbol{1}$ stands for the number of examples selected by the gate $\boldsymbol{\gamma}$ (with $\bar{n}_1 < n$).

When looking at the bias terms, we see how Pi-DUAL can compensate for a larger variance term to achieve a lower risk overall. We first recall the definition of $\boldsymbol{\delta}^\star = \boldsymbol{X}\boldsymbol{w}^\star - \boldsymbol{A}\boldsymbol{v}^\star$ that computes the difference between the contributions of the standard features $\boldsymbol{X}$ and the PI features $\boldsymbol{A}$. If the level of noise explained by $\boldsymbol{A}_2$ has a large contribution compared with the signal from $\boldsymbol{X}_2$, the second part $\boldsymbol{\delta}_2^\star$ of $\boldsymbol{\delta}^\star$ can contain large entries. While $\boldsymbol{w}_0$ has a bias term scaling with $\mathcal{O}((\boldsymbol{I} - \boldsymbol{\gamma}^*)\boldsymbol{\delta}^\star)$—that is, proportional to the number $n_2$ of noisy examples captured by $\boldsymbol{\delta}_2^\star$—we can observe that $\boldsymbol{w}_1$ has a more robust scaling. Indeed, it depends on $\mathcal{O}(\boldsymbol{\Delta}_\gamma \boldsymbol{\delta}^\star)$ that only scales with the number of disagreements between the reference gate $\boldsymbol{\gamma}^\star$ and that used for training $\boldsymbol{\gamma}$.

## A.3 Proof: Risk of ordinary least squares

We assume that $\boldsymbol{Q}$ is invertible. We focus on the solution of

$$\min_{\boldsymbol{w} \in \mathbb{R}^d} \frac{1}{2n}\|\boldsymbol{y} - \boldsymbol{X}\boldsymbol{w}\|^2 \tag{8}$$

that is given by

$$\begin{aligned} \boldsymbol{w}_0 &= \boldsymbol{Q}^{-1}\boldsymbol{X}^\top \boldsymbol{y} \\ &= \boldsymbol{Q}^{-1}\boldsymbol{X}^\top (\boldsymbol{\gamma}^* \boldsymbol{X}\boldsymbol{w}^\star + (\boldsymbol{I} - \boldsymbol{\gamma}^*)\boldsymbol{A}\boldsymbol{v}^\star + \varepsilon) \\ &= \boldsymbol{Q}^{-1}\boldsymbol{X}^\top (-(\boldsymbol{I} - \boldsymbol{\gamma}^*)\boldsymbol{\delta}^\star + \boldsymbol{X}\boldsymbol{w}^\star + \varepsilon) \\ &= -\boldsymbol{Q}^{-1}\boldsymbol{X}^\top (\boldsymbol{I} - \boldsymbol{\gamma}^*)\boldsymbol{\delta}^\star + \boldsymbol{w}^\star + \boldsymbol{Q}^{-1}\boldsymbol{X}^\top \varepsilon. \end{aligned}$$

Plugging into Eq. (5), we obtain

$$\mathcal{R}(\boldsymbol{w}_0) = \frac{1}{n_1}\|\boldsymbol{\gamma}^\star \boldsymbol{\Pi}_x (\boldsymbol{I} - \boldsymbol{\gamma}^*)\boldsymbol{\delta}^\star - \boldsymbol{\gamma}^\star \boldsymbol{\Pi}_x \varepsilon\|^2 + \sigma^2.$$

Expanding the square and using that $\text{tr}(\gamma^*\mathbf{\Pi}_x(\gamma^*\mathbf{\Pi}_x)^\top) = \text{tr}(\gamma^*\mathbf{\Pi}_x) = \text{tr}(\mathbf{Q}^{-1}\mathbf{Q}_1)$, the final risk expression is

$$
\begin{aligned}
\mathbb{E}[\mathcal{R}(\boldsymbol{w}_0)] &= \frac{1}{n_1}\|\gamma^\star\mathbf{\Pi}_x(\boldsymbol{I} - \gamma^*)\boldsymbol{\delta}^\star\|^2 + \frac{1}{n_1}\mathbb{E}[\|\gamma^\star\mathbf{\Pi}_x\varepsilon\|^2] + \sigma^2 \\
&= \frac{1}{n_1}\|\gamma^\star\mathbf{\Pi}_x(\boldsymbol{I} - \gamma^*)\boldsymbol{\delta}^\star\|^2 + \frac{\sigma^2}{n_1}\text{tr}(\boldsymbol{Q}^{-1}\boldsymbol{Q}_1) + \sigma^2.
\end{aligned}
\tag{9}
$$

## A.4 PROOF: RISK OF PI-DUAL

We focus on the solution of

$$
\min_{\boldsymbol{w}\in\mathbb{R}^d, \boldsymbol{v}\in\mathbb{R}^m} \frac{1}{2n}\|\boldsymbol{y} - (\gamma\boldsymbol{X}\boldsymbol{w} + (\boldsymbol{I} - \gamma)\boldsymbol{A}\boldsymbol{v})\|^2
\tag{10}
$$

to construct an estimator. Here, $\gamma$ refers to a diagonal mask matrix of size $n \times n$ which we use as (pre-defined) gates for Pi-DUAL. We introduce the notations:

- The masked versions of $\boldsymbol{X}$ and $\boldsymbol{A}$: $\bar{\boldsymbol{X}} = \gamma\boldsymbol{X}$ and $\bar{\boldsymbol{A}} = (\boldsymbol{I} - \gamma)\boldsymbol{A}$
- The projector onto the span of the columns of $\bar{\boldsymbol{A}}$:

$$
\bar{\mathbf{\Pi}}_a = \bar{\boldsymbol{A}}(\bar{\boldsymbol{A}}^\top\bar{\boldsymbol{A}})^{-1}\bar{\boldsymbol{A}}^\top \in \mathbb{R}^{n\times n}
$$

- The projection $\bar{\boldsymbol{X}}_a = (\boldsymbol{I} - \bar{\mathbf{\Pi}}_a)\bar{\boldsymbol{X}} \in \mathbb{R}^{n\times d}$ of $\bar{\boldsymbol{X}}$ onto the orthogonal of the span of the columns of $\bar{\boldsymbol{A}}$, and the matrices

$$
\bar{\boldsymbol{H}} = (\bar{\boldsymbol{X}}_a^\top\bar{\boldsymbol{X}}_a)^{-1}\bar{\boldsymbol{X}}_a^\top \in \mathbb{R}^{d\times n} \text{ and } \bar{\boldsymbol{Q}}_a = \bar{\boldsymbol{X}}_a^\top\bar{\boldsymbol{X}}_a \in \mathbb{R}^{d\times d}.
$$

We can reuse Lemma I.3 from Collier et al. (2022), with $(\bar{\boldsymbol{X}}, \bar{\boldsymbol{A}})$ in lieu of $(\boldsymbol{X}, \boldsymbol{A})$. The solution of $\boldsymbol{w}$ is thus given by

$$
\begin{aligned}
\boldsymbol{w}_1 &= \bar{\boldsymbol{H}}\boldsymbol{y} \\
&= \bar{\boldsymbol{H}}(\gamma^*\boldsymbol{X}\boldsymbol{w}^\star + (\boldsymbol{I} - \gamma^*)\boldsymbol{A}\boldsymbol{v}^\star + \varepsilon) \\
&= \bar{\boldsymbol{H}}((\boldsymbol{\Delta}_\gamma + \gamma)\boldsymbol{X}\boldsymbol{w}^\star + (\boldsymbol{I} - (\boldsymbol{\Delta}_\gamma + \gamma))\boldsymbol{A}\boldsymbol{v}^\star + \varepsilon) \\
&= \bar{\boldsymbol{H}}(\boldsymbol{\Delta}_\gamma\boldsymbol{\delta}^\star + \bar{\boldsymbol{X}}\boldsymbol{w}^\star + \bar{\boldsymbol{A}}\boldsymbol{v}^\star + \varepsilon) \\
&= \bar{\boldsymbol{H}}\boldsymbol{\Delta}_\gamma\boldsymbol{\delta}^\star + \boldsymbol{w}^\star + \boldsymbol{0} + \bar{\boldsymbol{H}}\varepsilon.
\end{aligned}
$$

where in the last line, we have used that $\bar{\boldsymbol{H}}\bar{\boldsymbol{X}} = (\bar{\boldsymbol{X}}_a^\top\bar{\boldsymbol{X}}_a)^{-1}\bar{\boldsymbol{X}}_a^\top\bar{\boldsymbol{X}}_a = \boldsymbol{I}$ (because $\boldsymbol{I} - \bar{\mathbf{\Pi}}_a = (\boldsymbol{I} - \bar{\mathbf{\Pi}}_a)^2$) and $(\boldsymbol{I} - \bar{\mathbf{\Pi}}_a)\bar{\boldsymbol{A}} = \boldsymbol{0}$.

Plugging into Eq. (5), we obtain

$$
\mathcal{R}(\boldsymbol{w}_1) = \frac{1}{n_1}\|\gamma^\star\boldsymbol{X}\bar{\boldsymbol{H}}\boldsymbol{\Delta}_\gamma\boldsymbol{\delta}^\star + \gamma^\star\boldsymbol{X}\bar{\boldsymbol{H}}\varepsilon\|^2 + \sigma^2.
$$

Expanding the square and using that $\text{tr}(\gamma^\star\boldsymbol{X}\bar{\boldsymbol{H}}(\gamma^\star\boldsymbol{X}\bar{\boldsymbol{H}})^\top) = \text{tr}(\gamma^\star\boldsymbol{X}\bar{\boldsymbol{Q}}_a^{-1}\boldsymbol{X}^\top\gamma^\star) = \text{tr}(\bar{\boldsymbol{Q}}_a^{-1}\boldsymbol{Q}_1)$, the final risk expression is

$$
\begin{aligned}
\mathbb{E}[\mathcal{R}(\boldsymbol{w}_1)] &= \frac{1}{n_1}\|\gamma^\star\boldsymbol{X}\bar{\boldsymbol{H}}\boldsymbol{\Delta}_\gamma\boldsymbol{\delta}^\star\|^2 + \frac{1}{n_1}\mathbb{E}[\|\gamma^\star\boldsymbol{X}\bar{\boldsymbol{H}}\varepsilon\|^2] + \sigma^2 \\
&= \frac{1}{n_1}\|\gamma^\star\boldsymbol{X}\bar{\boldsymbol{H}}\boldsymbol{\Delta}_\gamma\boldsymbol{\delta}^\star\|^2 + \frac{\sigma^2}{n_1}\text{tr}(\bar{\boldsymbol{Q}}_a^{-1}\boldsymbol{Q}_1) + \sigma^2.
\end{aligned}
\tag{11}
$$

# B  EXPERIMENTAL DETAILS

We now report the main details of all our experiments. All our experiments, including the reimplementation of other noisy label methods, are built on the open-source `uncertainty_baselines` codebase (Nado et al., 2021) and follow as much as possible the benchmarking practices of Ortiz-Jimenez et al. (2023).

## B.1  DATASETS

We use the following PI datasets to evaluate the performance of Pi-DUAL and other methods:

**CIFAR-10H** (Peterson et al., 2019) is a relabeled version for CIFAR-10 (Krizhevsky et al., 2009) test set with 10,000 images. However, as proposed by Collier et al. (2022) we use CIFAR-10H as a training set so we use the standard CIFAR-10 training set as our test set. Following Ortiz-Jimenez et al. (2023), we use the noisiest version of CIFAR-10H (denoted as "worst") in our experiments. It has a noise rate (defined as the percentage of the labels that disagree with the original CIFAR-10 dataset) of approximately 64.6%. The PI of CIFAR-10H consists of annotator IDs, annotator experiences and the time taken for the annotations.

**CIFAR-10N and CIFAR-100N** (Wei et al., 2022) are relabeled versions of CIFAR-10 and CIFAR-100 with noisy human annotations. In our experiments, we use the noisiest version of these two datasets, known as CIFAR-10N (worst) and CIFAR-100N (fine), which both have a 40.2% noise rate. The PI on these datasets consist on annotator IDs and annotator experience. It is worth noting that compared to CIFAR-10H, and as reported by Ortiz-Jimenez et al. (2023), the PI on these two datasets is of a much lower quality. In general, it is much less predictive of the presence of a label mistake on a specific sample, as the PI features are only provided as averages over batches of samples.

**ImageNet-PI** (Ortiz-Jimenez et al., 2023) is a relabeled version of the ImageNet ILSVRC12 dataset (Deng et al., 2009). In contrast to the human-relabeled datasets described above, the labels of ImageNet-PI are provided by 16 different deep neural networks pre-trained on the original ImageNet. The PI for this dataset contains the annotator confidence, the annotator ID, the number of parameters of the model and its accuracy. In our experiments, we use both the high-noise (83.8% noise rate) and low-noise version (48.1% noise rate) of ImageNet-PI.

A summary of the features of these datasets is given in Tab. 5.

Table 5: Summary of the main features of each of the datasets used in our experiments.

|                   | CIFAR-10H (worst) | CIFAR-10N (worst) | CIFAR-100N (fine) | ImageNet-PI (low-noise) | ImageNet-PI (high-noise) |
|-------------------|-------------------|-------------------|-------------------|-------------------------|--------------------------|
| Training set size | 10k               | 50k               | 50k               | 1.28M                   | 1.28M                    |
| PI quality        | High              | Low               | Low               | High                    | High                     |
| Annotators        | Humans            | Humans            | Humans            | Models                  | Models                   |
| Noise rate        | 64.6%             | 40.2%             | 40.2%             | 48.1%                   | 83.8%                    |

## B.2  BASELINES

In our experiments, we compare the performance of Pi-DUAL on different tasks against several baselines selected to provide a fair comparison and good coverage of different methods in the literature. Specifically, we restrict ourselves to methods that only require training a single model on a single stage. We discard, therefore, methods that need multiple stages of training, such as Forward-T (Patrini et al., 2017) or Distillation PI (Lopez-Paz et al., 2015); or multiple models, such as co-teaching (Han et al., 2018) and DivideMix (Li et al., 2020b), as these are more computationally demanding, harder to tune, and in general harder to scale to the large-scale settings we are interested in. Also, most of these methods have been compared against more recent strategies like SOP (Liu et al., 2022) or ELR (Liu et al., 2020), and shown to perform worse than these baselines.

A short description of each of the baselines we compare to is provided below:

**Cross-entropy** Conventional training strategy consisting in the direct minimization of the cross-entropy loss between the model's predictions and the noisy labels.

**SOP** (Liu et al., 2022) A noise-modeling method which models the label noise as an additive sparse signal. During training SOP uses the implicit bias of a custom overparameterized formulation to drive the learning of this sparse components. SOP needs $\mathcal{O}(n \times K)$ extra parameters over the cross-entropy baseline, where $n$ is the number of training samples, and $K$ the number of training classes.

**ELR** (Liu et al., 2020) This method adds an extra regularization term to the cross-entropy loss to bias the model's predictions towards their value in the early stages of training. To that end, it requires storing a moving average of the model predictions at each training iteration which adds $\mathcal{O}(n \times K)$ extra parameters over the cross-entropy baseline.

**HET** (Collier et al., 2021) Another noise-modeling method which models the uncertainty in the predictions as heteroscedastic per-sample Gaussian component in the logit space. The original version scales poorly with the number of classes, but the more recent HET-XL version (Collier et al., 2023), allows to scale this modeling approach to datasets with thousands of classes with only $\mathcal{O}(1)$ extra parameters coming from the small network that parameterizes the covariance of the noise. The standard version of HET achieves similar performance to HET-XL on ImageNet and can be run efficiently on this dataset. In our experiments, we thus use HET instead of HET-XL as a baseline.

**TRAM** (Collier et al., 2022) A PI-method which uses two heads, one with access to PI and one without it, to learn $p(\tilde{y}|\boldsymbol{x}, \boldsymbol{a})$ and $p(\tilde{y}|\boldsymbol{x})$ respectively. However, the feature extraction network leading to these heads is only trained using the gradients coming from the PI head. During inference, only the no-PI head is used. TRAM only requires $\mathcal{O}(1)$ extra parameters for the additional PI head.

**TRAM++** (Ortiz-Jimenez et al., 2023) On top of TRAM, TRAM++ augments the PI features with random sample-identifier to encourage the model to use the PI as a learning shortcut to memorize the noisy labels.

**AFM** (Collier et al., 2022) Another PI-method that during training learns to approximate $p(\tilde{y}|\boldsymbol{x}, \boldsymbol{a})$ and during inference uses approximate marginalization based on the independence assumption $p(\boldsymbol{a}|\boldsymbol{x}) \approx p(\boldsymbol{a})$ and Monte-Carlo sampling to marginalize over $\boldsymbol{a}$. AFM only requires $\mathcal{O}(1)$ extra parameters to accomodate for the PI in the last layers.

### B.3 HYPERPARAMETER TUNING STRATEGY

As mentioned in Sec. 4, to ensure a fair comparison of the different methods, we apply the same hyperparameter tuning strategy in all our experiments and for all methods. In particular, we use a noisy validation set taken from the training set to select the best hyperparameters of a grid search. On CIFAR-10H, we randomly select $4\%$ of the samples; on CIFAR-10N and CIFAR-100N, $2\%$; and on ImageNet-PI, $1\%$ of all the samples in the training set. In all the experiments presented in the main text we use early stopping to select the best epoch to evaluate each method. Early stopping is also performed over the noisy validation set, although the reported accuracies are given over the clean test set.

### B.4 TRAINING DETAILS FOR CIFAR

**General settings** We use a WideResNet-10-28 architecture in our CIFAR experiments. We train all models for 90 epochs, with the learning rate decaying multiplicatively by $0.2$ after 36, 72 and 96 epochs. We use a batch size of 256 in all experiments, and train the models with an SGD optimizer with 0.9 Nesterov momentum. In our grid searches, we sweep over the initial learning rate $\{0.01, 0.1\}$ and weight decay strength $\{10^{-4}, 10^{-3}\}$. We always we use random crops combined with random horizontal flips as data augmentation.

**Method-specific settings** For ELR (Liu et al., 2020), we additionally sweep over the temporal ensembling parameter $\beta$ of $\{0.5, 0.7, 0.9\}$ and the regularization coefficient $\lambda$ of $\{1, 3, 7\}$. For SOP (Liu et al., 2022), we sweep over the learning rate for $u_i$ of $\{1, 10, 100\}$, as well as the learning rate for $v_i$ of $\{1, 10, 100, 1000\}$. We refer to the original papers for ELR (Liu et al., 2020) and SOP (Liu et al., 2022) for detailed illustration of the hyperparameters.

For TRAM and AFM, we set the PI tower width to 1024 following the settings in Collier et al. (2022). For TRAM++, we tune the PI tower width over a range of {512, 1024, 2048, 4096}. Additionally, we tune the random PI length of {8, 14, 28}, and no-PI loss weight over {0.1, 0.5} for TRAM++.

For HET, we tune the heteroscedastic temperature over a range of {0.25, 0.5, 0.75, 1.0, 1.25, 1.5, 2.0, 3.0, 5.0}. For CIFAR-10H and CIFAR-10N, the number of factors for the low-rank component of the heteroscedastic covariance matrix is set to 3, while we set it to 6 for CIFAR-100N experiments.

For Pi-DUAL, we set the width of the noise network and gating network to 1024 in the CIFAR-10H experiments, and to 2048 for the experiments with CIFAR-10N and CIFAR-100N. We use a three-layer MLP with ReLU activations for the noise network. The gating network shares the first layer with the noise network followed by another two fully-connected layers with ReLU activations and a sigmoid activation at its output. We additionally search the random PI length over {4,8,12,16}. We do not apply weight decay regularization on the gating network and noise network for experiments on CIFAR for Pi-DUAL.

We highlight that, compared with the competing methods (except the cross-entropy baseline), Pi-DUAL requires the smallest budget for hyperparameter tuning as it requires to tune only one additional hyperparameter, i.e., the random PI length; while the other methods require tuning at least two additional method-specific hyperparameters.

### B.5 TRAINING DETAILS FOR IMAGENET-PI

**General settings.** For all experiments with ImageNet-PI, we use a ResNet-50 architecture and SGD optimizer with Nesterov momentum of 0.9. The models are trained for 90 epochs in total with a batch size of 2048, with the learning rate decaying multiplicatively by 0.1 after 30, 60 and 80 epochs. The initial learning rate is set to 0.1, and we search over $\{10^{-5}, 10^{-4}\}$ for the weight decay strength. Random crop and random horizontal flip are used for data augmentation.

**Method-specific settings.** For all PI-related baselines (TRAM, TRAM++, AFM), we set the PI tower width to 2048. We set the no-PI loss weight to 0.5 and set the random PI length of 30 for TRAM++. For HET, we set the number of factors for the low-rank component of the heteroscedastic covariance matrix to 15, and we set the heteroscedastic temperature to 3.0.

For Pi-DUAL, we set the random PI length to 30. The weight decay regularization on the gating network and noise network is the same as the prediction network. The architecture of the noise and gating network is the same as the one of the CIFAR experiments with a width of 2048.

# C  ADDITIONAL RESULTS

## C.1  TRAINING DYNAMICS ON CIFAR-10N AND IMAGENET-PI (LOW NOISE)

Here we provide the training dynamics for CIFAR-10N and ImageNet-PI (low noise) in Fig. 5 with the same findings as in Sec. 5.1.

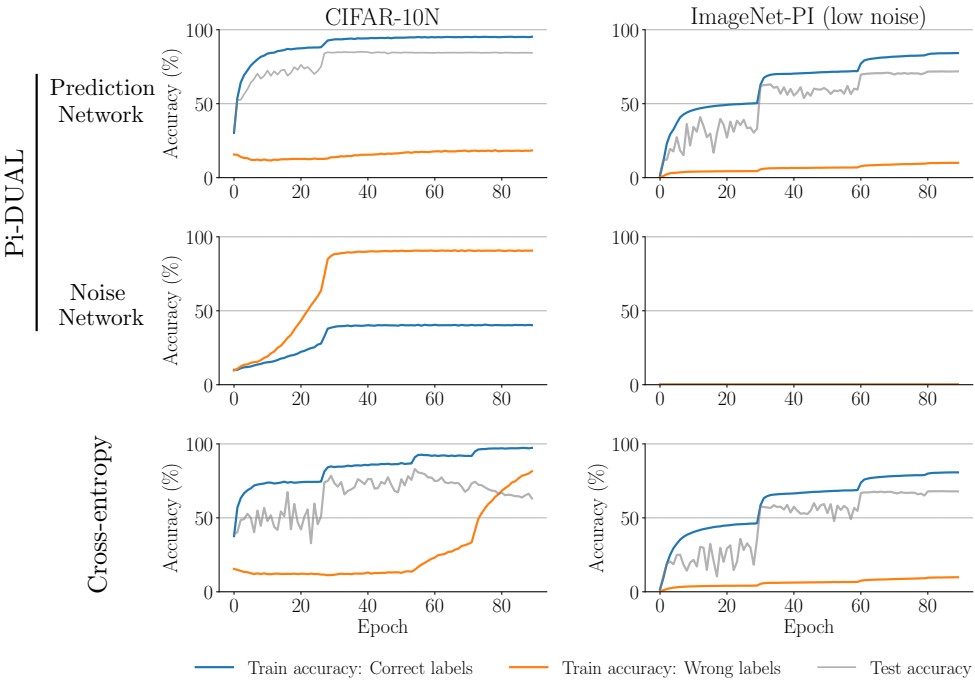

Figure 5: Training accuracy dynamics on correct and mislabled samples, respectively for the prediction and noise sub-networks on CIFAR-10N (worst) and ImageNet-PI (low noise).

## C.2  DISTRIBUTION OF THE GATING NETWORK PREDICTIONS ON CIFAR-10N AND IMAGENET-PI (LOW NOISE)

We show the distribution of the predictions of the gating network for CIFAR-10N and ImageNet-PI (low noise) in Fig. 6 with the same findings as in Sec. 5.2.

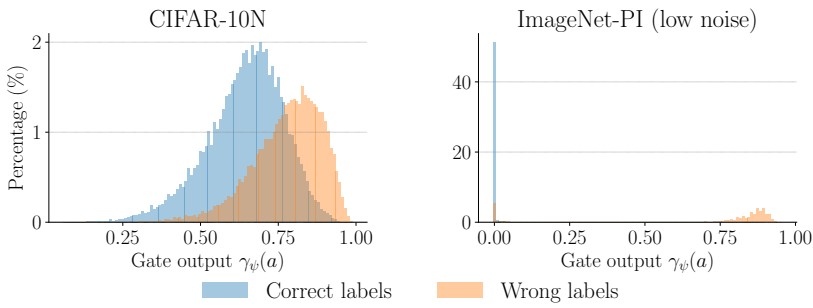

Figure 6: Distribution of $\gamma_\psi(a)$, on correct training examples with correct and wrong labels for CIFAR-10N and ImageNet-PI (low noise).

## C.3 DISTRIBUTION OF THE PREDICTION NETWORK CONFIDENCE

We present the distribution of the prediction confidence of the prediction network on observed labels for all 5 datasets in Fig. 7 complementing the findings of Sec. 4.3. From the figure, we see that the confidence of the prediction network is clearly separated over samples with clean and wrong labels.

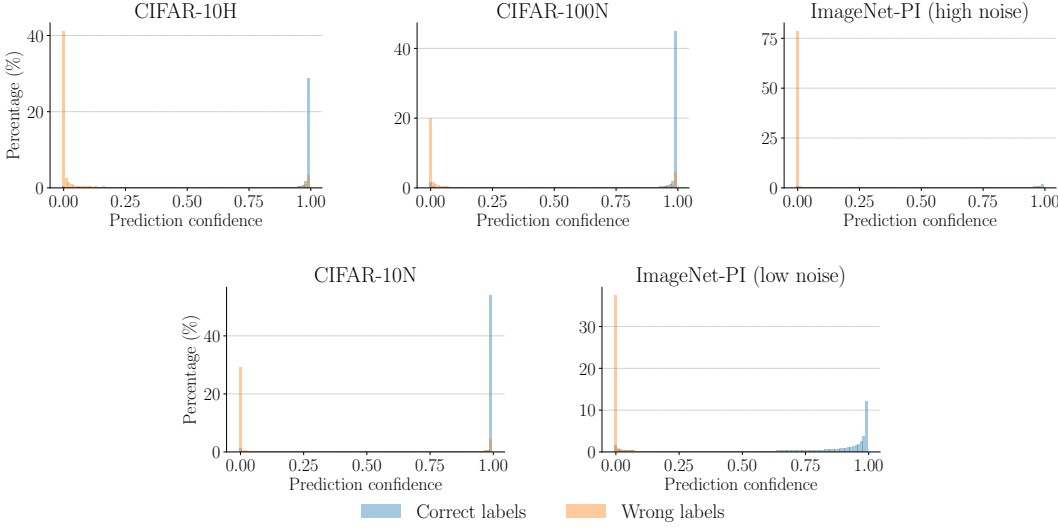

Figure 7: Distribution for the generalization network's confidence on the noisy labels, separated by correct and wrong samples.

## C.4 RESULTS WITHOUT EARLY-STOPPING

In the main text, as it is standard practice in the literature, we always provided results using early stopping. However, as mentioned before, early stopping is a key ingredient to achieve good performance by other methods, not Pi-DUAL. Indeed, Pi-DUAL still barely overfits to the incorrect labels using the prediction network, and thus it does not require early-stopping to achieve good results. To demonstrate this, Tab. 6 reports the results of the same experiments as in Tab. 2, but without using early stopping. From the table, we observe the performance of Pi-DUAL does not degrade in any of the datasets, while for other methods it suffers a heavily.

Table 6: Test accuracy (without early stopping) on CIFAR-10H, CIFAR-10N, CIFAR-100N, and ImageNet-PI, comparing Pi-DUAL with previous methods (grouped by PI-based methods and No-PI methods). Mean and standard deviation are reported over 5 individual runs for CIFAR experiments and 3 runs for ImageNet experiments.

| | Methods | CIFAR-10H (worst) | CIFAR-10N (worst) | CIFAR-100N (fine) | ImageNet-PI (low-noise) | ImageNet-PI (high-noise) |
|---|---|---|---|---|---|---|
| No-PI | Cross-entropy | $42.4_{\pm0.2}$ | $67.7_{\pm0.6}$ | $55.8_{\pm0.2}$ | $68.2_{\pm0.2}$ | $47.4_{\pm0.4}$ |
| | ELR | $49.2_{\pm1.2}$ | $84.3_{\pm0.4}$ | $\mathbf{63.8}_{\pm0.3}$ | - | - |
| | SOP | $50.3_{\pm0.7}$ | $\mathbf{86.0}_{\pm0.3}$ | $61.1_{\pm0.2}$ | - | - |
| PI | TRAM | $59.2_{\pm0.2}$ | $67.0_{\pm0.4}$ | $56.4_{\pm0.3}$ | $69.6_{\pm0.1}$ | $54.0_{\pm0.0}$ |
| | TRAM++ | $64.7_{\pm0.6}$ | $82.3_{\pm0.1}$ | $60.6_{\pm0.2}$ | $69.5_{\pm0.0}$ | $54.1_{\pm0.1}$ |
| | AFM | $61.2_{\pm0.7}$ | $69.8_{\pm0.5}$ | $58.9_{\pm0.3}$ | $70.3_{\pm0.0}$ | $55.3_{\pm0.2}$ |
| | Pi-DUAL (Ours) | $\mathbf{73.8}_{\pm0.3}$ | $84.9_{\pm0.3}$ | $\mathbf{64.2}_{\pm0.3}$ | $\mathbf{71.7}_{\pm0.1}$ | $\mathbf{62.3}_{\pm0.1}$ |

The same applies in the case of the noise detection results, where in Tab. 7 we see that Pi-DUAL can still detect the noisy labels equally well as in Tab. 3 without the use of early stopping. The other methods on the other hand perform worse when applied to the last training epoch than to the early stopped one.

Table 7: AUC of different noise detection methods without using early-stopping.

| Methods | CIFAR-10H (worst) | CIFAR-10N (worst) | CIFAR-100N (fine) | ImageNet-PI (low-noise) | ImageNet-PI (high-noise) |
|---|---|---|---|---|---|
| Cross-entropy | 0.558 | 0.676 | 0.666 | 0.935 | 0.941 |
| ELR | 0.660 | 0.839 | 0.843 | - | - |
| SOP | 0.743 | 0.932 | 0.793 | - | - |
| TRAM++ | 0.887 | 0.946 | 0.890 | 0.937 | 0.959 |
| Pi-DUAL (conf.) | 0.972 | **0.960** | **0.910** | **0.953** | **0.987** |
| Pi-DUAL (gate) | **0.983** | 0.815 | 0.729 | **0.953** | 0.986 |

## C.5 AUGMENTING Pi-DUAL WITH STATE-OF-THE-ART REGULARIZATION TECHNIQUES

Prior works in the literature have shown that the noisy-label training methods can be boosted with techniques from semi-supervised learning domain (Li et al., 2020a; Liu et al., 2020; 2022), while inevitably introducing extra complexity costs as well as more complexity. In this section, we propose an extension of Pi-DUAL, Pi-DUAL+, which adds to Pi-DUAL two regularization techniques, label smoothing and prediction consistency regularizer.

**Label smoothing** Label smoothing is a regularization technique that was introduced to mitigate overconfidence during training by replacing hard labels with smoothed soft labels (Szegedy et al., 2016). It has become a widely-used method to improve model generalization performance in classification tasks.

**Consistency regularizer** Prediction consistency regularizer is commonly used in both the semi-supervised learning (Berthelot et al., 2019; Sohn et al., 2020; Xie et al., 2020) and learning with label noise literature (Cheng et al., 2020; Liu et al., 2022). It encourages the prediction consistency of the model across different input views. In Pi-DUAL+, we add a consistency regularizer $\mathcal{L}_C$ on the generalization term. Specifically, $\mathcal{L}_C$ is defined as the Kullback-Leibler divergence between softmax prediction from images with the default augmentation in Sec. B.4 and softmax predictions from the corresponding images augmented by Unsupervised Data Augmentation (Xie et al., 2020):

$$\mathcal{L}_C = \frac{1}{N} \sum_{i=1}^{N} D_{\mathrm{KL}}(\mathrm{softmax}(f_{\boldsymbol{\theta}}(\boldsymbol{x}_i)) \parallel \mathrm{softmax}(f_{\boldsymbol{\theta}}(\mathrm{UDA}(\boldsymbol{x}_i)))) \tag{12}$$

We use a hyper-parameter $\lambda_C$ to control the strength of the consistency regularizer.

For Pi-DUAL+, we sweep over the label smoothing over $\{0, 0.4\}$, and $\lambda_C$ over $\{0.5, 1\}$. We train the Pi-DUAL+ for 300 epochs with a batch size of 128. The learning rate is set as 0.1 and decays with a cosine annealing schedule (Loshchilov & Hutter, 2016). Additionally, we sweep over the random-pi length over $\{4, 8\}$ and set the l2 regularization strength to $1e^{-4}$.

We compare Pi-DUAL+ against several semi-supervised learning pipeline methods, including Divide-Mix Li et al. (2020b), CORES* (Cheng et al., 2020), PES(semi) (Bai et al., 2021), ELR+ (Liu et al., 2020) and SOP+ (Liu et al., 2022). The results are compared in three datasets, CIFAR-10H, CIFAR-10N and CIFAR-100N, as shown in Tab. C.5.

From the table, we observe that, Pi-DUAL+ outperforms the other methods by a margin of over 11.0 points on CIFAR-10H, while it performs on par with the state-of-the-art on the other two CIFAR-*N datasets. It demonstrates again the importance of good-quality PI features to maximize the performance of Pi-DUAL/Pi-DUAL+, and also demonstrates that Pi-DUAL can be effected boosted by semi-supervised learning methods.

## C.6 ABLATIONS OVER MODEL STRUCTURES

### C.6.1 ABLATION FOR PREDICTION NETWORK BACKBONE

In all our CIFAR-level experiments (for both Pi-DUAL and other baselines methods), we used a WideResNet-10-28 as model backbone. Here we replace the WideResNet-10-28 with a ResNet-34 and present the performance comparison between Pi-DUAL and CE baseline on Table C.6.1.

|  | CIFAR-10H | CIFAR-10N | CIFAR-100N |
|---|---|---|---|
| CE | $51.10_{\pm 2.20}$ | $80.60_{\pm 0.20}$ | $60.40_{\pm 0.50}$ |
| Divide-Mix | $71.68_{\pm 0.27}$ | $92.56_{\pm 0.42}$ | $\mathbf{71.13}_{\pm 0.48}$ |
| PES(semi) | $71.16_{\pm 1.78}$ | $92.68_{\pm 0.22}$ | $70.36_{\pm 0.33}$ |
| ELR+ | $54.46_{\pm 0.50}$ | $91.09_{\pm 1.60}$ | $66.72_{\pm 0.07}$ |
| CORES* | $57.80_{\pm 0.57}$ | $91.66_{\pm 0.09}$ | $55.72_{\pm 0.42}$ |
| SOP+ | $66.02_{\pm 0.06}$ | $\mathbf{93.24}_{\pm 0.21}$ | $67.81_{\pm 0.23}$ |
| Pi-DUAL+ | $\mathbf{83.23}_{\pm 0.26}$ | $93.31_{\pm 0.21}$ | $67.99_{\pm 0.08}$ |

Table 8: Test accuracy on CIFAR-10H, CIFAR-10N and CIFAR-100N, comparing Pi-DUAL+ against state-of-the-art methods which combine noisy labels techniques with semi-supervised learning methods. The results of baseline methods on CIFAR-10N and CIFAR-100N are taken from Wei et al. (2021); Liu et al. (2022)

|  | WideResNet-10-28 | | ResNet34 | |
|---|---|---|---|---|
| Dataset \Method | CE | Pi-DUAL | CE | Pi-DUAL |
| CIFAR-10H | $51.1_{\pm 2.2}$ | $\mathbf{71.3}_{\pm 3.3}$ | $51.4_{\pm 2.1}$ | $\mathbf{69.3}_{\pm 3.1}$ |
| CIFAR-10N | $80.6_{\pm 0.2}$ | $\mathbf{84.9}_{\pm 0.4}$ | $80.7_{\pm 1.0}$ | $\mathbf{84.5}_{\pm 0.4}$ |
| CIFAR-100N | $60.4_{\pm 0.5}$ | $\mathbf{64.2}_{\pm 0.3}$ | $56.9_{\pm 0.4}$ | $\mathbf{62.2}_{\pm 0.3}$ |

Table 9: Accuracy comparison between Pi-DUAL and CE on three datasets, using two different model backbones: WideResNet-10-28 and ResNet34.

From the results, we observe that Pi-DUAL maintains its performance improvement over CE with a different model backbone, exceeding the performance of CE baseline by a notable margin in all three datasets.

### C.6.2    ABLATIONS ON STRUCTURE FOR NOISE AND GATING NETWORKS

**Ablations on the width** In the paper we set the width of the PI-related modules (for both noise network and gating network) by default to 1024 for CIFAR-10H, and 2048 for all other experiments, without fine-tuning. Here we provide the results for using different widths for the PI networks on three CIFAR datasets in Table C.6.2.

| Dataset \Model width | 512 | 1024 | 2048 | 4096 |
|---|---|---|---|---|
| CIFAR-10H | $\mathbf{72.8}_{\pm 2.9}$ | $\mathbf{71.3}_{\pm 3.3}$ | $\mathbf{71.4}_{\pm 3.6}$ | $\mathbf{71.2}_{\pm 3.8}$ |
| CIFAR-10N | $83.8_{\pm 0.4}$ | $83.6_{\pm 0.7}$ | $\mathbf{84.9}_{\pm 0.4}$ | $\mathbf{85.3}_{\pm 0.2}$ |
| CIFAR-100N | $62.3_{\pm 0.2}$ | $63.7_{\pm 0.4}$ | $\mathbf{64.2}_{\pm 0.3}$ | $\mathbf{64.4}_{\pm 0.2}$ |

Table 10: Accuracy of Pi-DUAL on three datasets, varying the width of noise and gating networks of Pi-DUAL.

From the table we observe that the performance of Pi-DUAL benefits from larger network width for the PI-related modules.

**Ablations on the depth** In the paper we set the depth of the PI-related modules by default to 3 by default for all experiments, without fine-tuning. Here we provide the results for using different depths for the PI networks on three CIFAR datasets in Table C.6.2. Note that the width of the PI-related modules are set to the default value when tuning the depth.

From the table we observe that the depth of the PI-related modules have to be at least of 3 layers to maximize the performance of Pi-DUAL.

| Dataset \Model depth | 2 | 3 | 4 |
|---|---|---|---|
| CIFAR-10H | $68.9_{\pm 2.0}$ | $71.3_{\pm 3.3}$ | $70.1_{\pm 3.5}$ |
| CIFAR-10N | $83.4_{\pm 0.4}$ | $84.9_{\pm 0.4}$ | $85.2_{\pm 0.3}$ |
| CIFAR-100N | $59.4_{\pm 1.0}$ | $64.2_{\pm 0.3}$ | $64.1_{\pm 0.2}$ |

Table 11: Accuracy of Pi-DUAL on three datasets, varying the depth of noise and gating networks of Pi-DUAL.

### C.7 TRAINING PI-DUAL WITH LOSS FUNCTION OF TRAM

While both TRAM and Pi-DUAL utilize PI features to combat label noise, Pi-DUAL uses a much simpler loss function than TRAM, which uses a weighted combination of two functions to train its two heads (Collier et al., 2022). As an ablation how Pi-DUAL benefits from its simple loss function which prevents the prediction network from overfitting to label noise, here we present the results for Pi-DUAL if we train it with the loss function of TRAM in Table C.7.

| Dataset \Method | CE | Pi-DUAL | Pi-DUAL with TRAM loss |
|---|---|---|---|
| CIFAR-10H | $51.1_{\pm 2.2}$ | $71.3_{\pm 3.3}$ | $61.6_{\pm 4.8}$ |
| CIFAR-10N | $80.6_{\pm 0.2}$ | $84.9_{\pm 0.4}$ | $81.6_{\pm 0.9}$ |
| CIFAR-100N | $60.4_{\pm 0.5}$ | $64.2_{\pm 0.3}$ | $59.0_{\pm 0.6}$ |

Table 12: Ablation for training Pi-DUAL using the loss function of TRAM.

### C.8 PERFORMANCE OF PI-DUAL WITH CORRUPTED PI FEATURES

We have emphasized in the main text the importance of the quality of PI features to the performance of Pi-DUAL. In this section, we perform an ablation study where we gradually corrupt the PI features of CIFAR-10H, and train Pi-DUAL with the dataset with corrupted PI features.

For each experiment, we randomly corrupt the PI features of a percentage of samples in the train set, where the corrupted PI features will be replaced by randomly generated PI features. Specifically, we substitute the annotator ID by a new random integer, and we substitute all other continual PI features by a random Gaussian vector with the same mean and standard deviation as the distribution of those features in the training set. We vary gradually the percentage of corrupted samples and report the performance for Pi-DUAL trained correspondingly in Table C.8.

From the table, we observe that the accuracy of Pi-DUAL decreases as there are more noise in the PI features of the training set, which demonstrates again the importance for high-quality PI features to maximize the performance of Pi-DUAL.

## D COMPUTATIONAL COST IN LARGE-SCALE DATASETS

In this paper, we used a TPU V3 with 8 cores for expeirments on ImageNet-PI, and A100 (40G) for experiments on CIFAR.

Here we provide a computational cost analysis for Pi-DUAL on ImageNet-PI, comparing it with other baseline methods, with respect to both number of parameters and training time in Table D.

| PI Corruption percentage | No corrupt | 25% corrupt | 50% corrupt | 75% corrupt | 100% corrupt |
|---|---|---|---|---|---|
| Accuracy | $71.3_{\pm 3.3}$ | $66.4_{\pm 2.7}$ | $63.9_{\pm 0.7}$ | $52.6_{\pm 4.0}$ | $47.2_{\pm 3.7}$ |

Table 13: Performance of Pi-DUAL with different levels of corruption in the PI features for the train set.

Note that in the table we do not have run time for SOP and ELR as these two methods are very hard to scale to ImageNet-PI, requiring over 1 billion parameters.

|  | CE | TRAM++ | Pi-DUAL | HET | SOP | ELR |
|---|---|---|---|---|---|---|
| Number of parameters | 26M | 32M | 36M | 58M | >1B | >1B |
| Run time per step | 0.510s | 0.541s | 0.566s | 0.575s | - | - |

Table 14: Computational cost analysis on ImageNet-PI in terms of number of parameters and running time, comparing Pi-DUAL with baseline methods. We do not have results for SOP and ELR as they are very hard to scale to ImageNet-PI as they require more than a billion parameters.

From the table, we can see that Pi-DUAL is a scalable method with almost the same training time as the cross-entropy baseline. Importantly, its parameters does not scale with either number of classes or number of samples, making it scalable to very large datasets.