# OpenReview forum: "Pi-DUAL: Using privileged information to distinguish clean from noisy labels"
_ICLR.cc/2024/Conference — Submitted to ICLR 2024_

### Official Review · Reviewer_h1hP · 2023-11-01

**Soundness:** 3 good
**Presentation:** 3 good
**Contribution:** 2 fair
**Rating:** 5
**Confidence:** 4

**Summary:**

This paper introduces a dual structure for learning with noisy labels by separating the training process into regular feature learning and privileged information learning. The regular feature learning module is responsible for the final inference. The effectiveness of the algorithm was validated on three datasets: CIFAR=1-H, CIFAR-N, and ImageNet-PI.

**Strengths:**

1. The structure of this paper is clear, making it easy for readers to follow.
2. It's intriguing to note that the no-PI network of the Dual structure outperforms previous PI-related works.
3. The results presented in the paper attest to the efficacy of the proposed method.

**Weaknesses:**

1. While PI is a concept introduced in prior works, this article doesn't offer significant innovations to it. The paper points out that PI-based methods underperform compared to no-PI-based methods. However, it fails to delve deep into the underlying principles causing this discrepancy. The conclusions seem to be drawn mainly from some experimental verifications rather than in-depth analysis.

2. The experimental comparisons are not exhaustive. Several state-of-the-art methods mentioned in Table 1, such as dividemix, weren't comparatively analyzed in the experiments. Given that Pi-Dual incorporates additional information to tackle label noise, comparing it with the current best methods is crucial to gauge the algorithm's effectiveness.

3. Certain ablation studies were not conducted, like the choice of the model backbone and parameters of the additional PI-related modules.

**Questions:**

Please refer to the Weaknesses section.

---

> ### Author Response · Authors · 2023-11-17
> **Response to Reviewer h1hP (Part 1)**
>
> We thank the reviewer for taking the time to review our paper and for all the valuable feedback. We address their comments below:
>
> **Innovations to PI**
>
> We respectfully disagree with the reviewer in that the innovations of Pi-DUAL are minor. As appreciated by the other reviewers, the goal of our paper is to build a simple and effective method that requires the minimal changes to existing training pipelines. In this regard, while Pi-DUAL is not the first in utilizing PI features to combat label noise, Pi-DUAL provides a solution that is both simpler and more efficient. Compared to TRAM, it has the following key distinctions:
>
> 1. **Network structure**: TRAM uses one model but with two heads, one for training and one for testing. In contrast,  Pi-DUAL uses three distinct subnetworks with clear roles during training and inference: the prediction network for fitting clean target distribution, the noise network for explaining away label noise, and the gate network for identifying noisy samples. These three roles can not be accounted for in the architecture of TRAM.
> 2. **Learning objective**: While the learning objective of TRAM is a weighted sum of two objectives (one for fitting the noisy target distribution, and another for transferring the knowledge to the no-PI head), the learning objective of Pi-DUAL is a simple cross-entropy loss over the noisy targets. For a better illustration of the differences between Pi-DUAL and TRAM, we conducted an ablation which trains Pi-DUAL with the loss function of TRAM, and show that this produces much worse results. This experiment has been added to Section C.8 in Appendix.
>
> | Dataset \ Method |    CE    |  Pi-DUAL | Pi-DUAL with TRAM loss |
> |:----------------:|:--------:|:--------:|:----------------------:|
> |     CIFAR-10H    | 51.1±2.2 | 71.3±3.3 |        61.6±4.8        |
> |     CIFAR-10N    | 80.6±0.2 | 84.9±0.4 |        81.6±0.9        |
> |    CIFAR-100N    | 60.4±0.5 | 64.2±0.3 |        59.0±0.6        |
>
> 3. **Modeling of label noise**. TRAM does not explicitly model label noise. Instead, it learns the conditional distribution of noisy labels with respect to both x and a, to eventually predict with an implicit marginalization over a. On the other hand, Pi-DUAL explicitly models per-instance-level label noise in its architecture, which permits it to identify noisy labels and to effectively reduce the negative impacts from noisy labels. In contrast to TRAM, which performs approximate marginalization over the PI,  the prediction network of Pi-DUAL learns the conditional distribution of clean labels with respect to only x. With this explicit modeling of label noise, Pi-DUAL delivers not only better performance, but also a much better interpretability as to how PI helps explain away label noise (e.g., how the noise network overfits to the wrong labels in Figure 2, and how the gate network identifies wrong labels in Figure 3).
>
> With these non-trivial improvements, Pi-DUAL outperforms TRAM++ on all 5 benchmarks achieving up to 8.1 accuracy points more than TRAM++ on ImageNet-PI. Pi-DUAL also delivers much better performance in identifying label noise post-training due to the explicit modeling of label noise.

---

> ### Author Response · Authors · 2023-11-17
> **Response to Reviewer h1hP (Part 2)**
>
> **Reasons why PI-methods underperform No-PI methods**
>
> We appreciate the comment of the reviewer, but we note that the overall goal of our work is to provide a better approach to utilize PI features, and not a deep analysis of prior work. In the paper, we have provided ample empirical and theoretical evidence illustrating the behavior of Pi-DUAL and we have tried to be as thorough as possible in our ablations. Pi-DUAL closes the gap between PI and no-PI methods even in datasets with low-quality PI features, while exhibiting much better scalability to no-PI methods.
>
> However, to answer the reviewer, we would like to emphasize that one of our main findings is that the relative performance between PI and No-PI methods depends heavily on the quality of PI features. For example, CIFAR-10H has fine-grained PI features, and the performance of TRAM++ outperforms the No-PI methods by 15.5 points. On the other hand, the PI feature quality for CIFAR-10N and CIFAR-100N are much worse as they are averaged over a batch of samples, which significantly reduces the amount of information on each sample carried by PI.
>
> To quantitatively illustrate the importance of PI quality in the performance of PI methods, we provide a new experiment where we randomly corrupt a percentage of PI features in the train set and train Pi-DUAL with the corrupted PI features (details in Appendix C.9). We copy the results here for reference:
>
> | PI Corruption | No corrupt | 25% corrupt | 50% corrupt | 75% corrupt | 100% corrupt |
> |:-------------:|:----------:|:-----------:|:-----------:|:-----------:|:------------:|
> |    Accuracy   |  71.3±3.3  |   66.4±2.7  |   63.9±0.7  |   52.6±4.0  |   47.2±3.7   |
>
> As we can see, as the quality of the PI gets worse, the performance of Pi-DUAL degrades significantly, highlighting the importance that good PI has on these methods.
>
> **Comparison to state-of-the-art**
>
> For a fair comparison, we only included in Table 2 the methods which use only one model and that are trained for a single stage. We did not compare to methods like Divide-Mix, because it incorporates techniques from semi-supervised learning, which substantially increases the computational costs (e.g., it uses two networks, mixup augmentation, longer training epochs, etc.). It is worth noting that prior works in label noise literature also do not compare directly to DivideMix, but compare DivideMix to versions of their methods which use semi-supervised learning techniques. For example, the authors for ELR (Liu et al., 2020) compare DivideMix with ELR+, while the authors for SOP (Liu et al., 2022) compare DivideMix with SOP+ (rather than SOP). ELR+ and SOP+ are respectively more sophisticated versions of the original methods augmented with semi-supervised learning techniques, making the comparison more fair in terms of both costs and complexity.
>
> Therefore, in response to the reviewers request for a comparison to Divide-Mix and other semi-supervised methods, and for a full validation of our proposed method and a fair comparison, we follow the common practice, and propose an augmented version of Pi-DUAL, Pi-DUAL+, and compare it to other semi-supervised learning methods. Pi-DUAL+ adds to Pi-DUAL two regularization techniques, label smoothing and prediction consistency regularization from UDA (Xie et al., 2020), where the latter one was a common choice for augmenting label noise training methods (Cheng et al., 2020, Liu et al., 2022). We have added the details for implementation of Pi-DUAL+ in Appendix C.5, and copy the results for Pi-DUAL+ below for reference, comparing it with several prior semi-supervised pipelines for combating label noise:
>
> |                |  CIFAR-10H |  CIFAR-10N | CIFAR-100N |
> |----------------|:----------:|:----------:|:----------:|
> |       CE       | 51.10±2.20 | 80.60±0.20 | 60.40±0.50 |
> |   Divide-Mix   | 71.68±0.27 | 92.56±0.42 | 71.13±0.48 |
> |    PES(semi)   | 71.16±1.78 | 92.68±0.22 | 70.36±0.33 |
> |      ELR+      | 54.46±0.50 | 91.09±1.60 | 66.72±0.07 |
> |     CORES*     | 57.80±0.57 | 91.66±0.09 | 55.72±0.42 |
> |      SOP+      | 66.02±0.06 | 93.24±0.21 | 67.81±0.23 |
> | Ours: Pi-DUAL+ | 83.23±0.26 | 93.31±0.21 | 67.99±0.08 |
>
> In this table, the baseline results are obtained using the code of the original papers if they are not available in the [public leaderboard](http://www.noisylabels.com). It is worthwhile noting that, on CIFAR-10H, Pi-DUAL+ outperforms the rest of the methods by a margin of over 11%, while it performs on par with the state-of-the-art methods on CIFAR-10N and CIFAR-100N. Importantly, this is because Pi-DUAL+ benefits from the high quality PI features in CIFAR-10H, while for CIFAR-10N and CIFAR-100N, , the PI is of poor quality, which explains why Pi-DUAL+ does not have a performance boost that is as large as the one in the case for CIFAR-10H. We hope it can help in delivering a thorough comparison with the state-of-the-art.

---

> ### Author Response · Authors · 2023-11-17
> **Response to Reviewer h1hP (Part 3)**
>
> **Architectural ablation studies**
>
> We thank the reviewer for bringing the missing ablation studies on the model architectures to our attention, and we hope the following additional ablations help complete our understanding of the method.
>
> > *Ablations for the model backbones*
>
> In the paper we use a WideResNet-10-28 for all experiments on CIFAR. Here we provide the results when replacing the architecture with a ResNet-34 instead. ResNet-34 is a common choice for model backbones in CIFAR-level experiments in the noise label literature (Cheng et al., 2020, Liu et al., 2020, 2022). We have added this ablation in Appendix C. 6.1. The results using ResNet-34 comparing Pi-DUAL and CE baseline are reported in the following table:
>
> |                  | WideResNet-10-28 |          | ResNet-34 |          |
> |------------------|------------------|----------|:---------:|:--------:|
> | Dataset \ Method |        CE        |  Pi-DUAL |     CE    |  Pi-DUAL |
> |     CIFAR-10H    |     51.1±2.2     | 71.3±3.3 |  51.4±2.1 | 69.3±3.1 |
> |     CIFAR-10N    |     80.6±0.2     | 84.9±0.4 |  80.7±1.0 | 84.5±0.4 |
> |    CIFAR-100N    |     60.4±0.5     | 64.2±0.3 |  56.9±0.4 | 62.2±0.3 |
>
> From the table, we observe that Pi-DUAL maintains its performance improvement over CE with a different model backbone, exceeding the performance of CE baseline by a notable margin in all three datasets.
>
> > *Ablations for PI network structure*
>
> Following the reviewer’s advice, we have added ablations on the width and depth of PI-related modules in Appendix C. 6.2.
>
> 1. *Ablations for the width for PI-related modules*: In the paper we set the width of the PI-related modules (for both noise network and gate network) by default to 1024 for CIFAR-10H, and 2048 for all other experiments, without tuning. Here we provide the results for different widths for the PI networks on three CIFAR datasets.
> | Dataset \ Width |    512   |   1024   |   2048   |   4096   |
> |:---------------:|:--------:|:--------:|:--------:|:--------:|
> |    CIFAR-10H    | 72.8±2.9 | 71.3±3.3 | 71.4±3.6 | 71.2±3.8 |
> |    CIFAR-10N    | 83.8±0.4 | 83.6±0.7 | 84.9±0.4 | 85.3±0.2 |
> |    CIFAR-100N   | 62.3±0.2 | 63.7±0.4 | 64.2±0.3 | 64.4±0.2 |
>
> From the table we observe that the performance of Pi-DUAL benefits from larger network width for the PI-related modules
>
> 2. *Ablations for the depth for PI-related modules*: In the paper we set the depth of the PI-related modules to 3 by default for all experiments, without tuning. Here we provide the results for different depths for the PI networks on three CIFAR datasets. Note that the width of the PI-related modules are set to the default value when tuning the depth.
> | Dataset \ Depth |     2    |     3    |     4    |
> |:---------------:|:--------:|:--------:|:--------:|
> |    CIFAR-10H    | 68.9±2.0 | 71.3±3.3 | 70.1±3.5 |
> |    CIFAR-10N    | 83.4±0.4 | 84.9±0.4 | 85.2±0.3 |
> |    CIFAR-100N   | 59.4±1.0 | 64.2±0.3 | 64.1±0.2 |
>
> From the table we observe that the depth of the PI-related modules have to be at least of 3 layers to maximize the performance of Pi-DUAL. We thank the reviewers for leading us to these interesting findings.

---

### Official Review · Reviewer_VkrZ · 2023-11-01

**Soundness:** 3 good
**Presentation:** 3 good
**Contribution:** 2 fair
**Rating:** 6
**Confidence:** 4

**Summary:**

This paper introduces PI-DUAL, a novel approach designed to address label noise by leveraging privileged information (PI). Privileged information is only accessible during the training phase. In essence, PI-DUAL employs neural networks to separately model the features of $x$ and the privileged information $a$, and it employs an additional network to determine the weights for the logits of $x$ and $a$.

To demonstrate the advantages of PI-DUAL, the authors conduct a series of comprehensive studies to investigate the training dynamics of PI-DUAL and to provide valuable theoretical insights. The proposed method is characterized by its simplicity, and experimental results exhibit promising performance when compared to other baseline methods."

**Strengths:**

- The method is both simple and technically sound. Notably, it represents the first PI-based method to explicitly model label noise.

- The studies conducted on PI-DUAL are thorough. The authors perform a range of experiments, including an exploration of training dynamics, an evaluation of detection performance, and an investigation of the impact of PI length. These experiments collectively contribute to a comprehensive assessment of the effectiveness of PI-DUAL.

- While the theoretical insights are based on the linear layer scenario, their analyses appear to be reasonable.

**Weaknesses:**

- The primary distinction between PI-DUAL and TRAM [R1] lies in PI-DUAL's approach of separately modeling features and privileged information (PI), as opposed to using two heads on top of the feature vector $x$. While this approach provides some technical novelty, it may be considered somewhat limited.

- I find the experimental results to be somewhat perplexing. It is reasonable that PI-DUAL doesn't need to be compared to two-stage methods like DivideMix or other semi-supervised pipelines. However, the test accuracy presented in Table 2 exhibits a notable gap compared to the results reported in the original papers or public leaderboards [R2]. Additionally, the test accuracy for TRAM on CIFAR-10N, as presented in the original paper, was 71.8, but in Table 2, it's only 64.9. While I acknowledge that differences in training settings may account for this, it would be beneficial to conduct specific experiments to compare each method under their optimal configurations.

- Typos:
    - In Section 3.2: Change "Here, $\gamma_{\phi}$ denotes" to "Here, $\gamma_{\psi}$ denotes."

[R1] Transfer and Marginalize: Explaining Away Label Noise with Privileged Information

[R2] You can access the relevant information at http://www.noisylabels.com.

**Questions:**

See *Weaknesses above*

---

> ### Author Response · Authors · 2023-11-17
> **Response to Reviewer VkrZ (Part 1)**
>
> We thank the reviewer for taking the time to review our paper and for all the valuable feedback. We address their comments below:
>
> **Differences between Pi-DUAL and TRAM**
>
> We respectfully disagree with the reviewer in that the innovations of Pi-DUAL with respect to TRAM are minor and correspond only to a change in architecture. As appreciated by this reviewer, the goal of our paper is to build a simple and effective method that requires minimal changes to existing training pipelines. In this regard, while both TRAM and Pi-DUAL utilize PI features to combat label noise, Pi-DUAL provides a solution that is both simpler and more efficient. Compared to TRAM, it has the following key distinctions:
>
> 1. **Network structure**: TRAM uses one model but with two heads, one for training and one for testing. In contrast,  Pi-DUAL uses three distinct subnetworks with clear roles during training and inference: the prediction network for fitting clean target distribution, the noise network for explaining away label noise, and the gate network for identifying noisy samples. These three roles can not be accounted for in the architecture of TRAM.
> 2. **Learning objective**: While the learning objective of TRAM is a weighted sum of two objectives (one for fitting the noisy target distribution, and another for transferring the knowledge to the no-PI head), the learning objective of Pi-DUAL is a simple cross-entropy loss over the noisy targets. For a better illustration of the differences between Pi-DUAL and TRAM, we conducted an ablation which trains Pi-DUAL with the loss function of TRAM, and show that this produces much worse results:
>
> | Dataset \ Method |    CE    |  Pi-DUAL | Pi-DUAL with TRAM loss |
> |:----------------:|:--------:|:--------:|:----------------------:|
> |     CIFAR-10H    | 51.1±2.2 | 71.3±3.3 |        61.6±4.8        |
> |     CIFAR-10N    | 80.6±0.2 | 84.9±0.4 |        81.6±0.9        |
> |    CIFAR-100N    | 60.4±0.5 | 64.2±0.3 |        59.0±0.6        |
>
> 3. **Modeling of label noise**. TRAM does not explicitly model label noise. Instead,  it learns the conditional distribution of noisy labels with respect to both x and a, to eventually predict with an implicit marginalization over a. On the other hand, Pi-DUAL explicitly models per-instance-level label noise in its architecture, which permits it to identify noisy labels and to effectively reduce the negative impact from noisy labels. In contrast to TRAM, the prediction network of Pi-DUAL learns the conditional distribution of clean labels with respect to x alone. With this explicit modeling of label noise, Pi-DUAL delivers not only better performance, but also a much better interpretability as to how PI helps explain away label noise (e.g., how the noise network overfits to the wrong labels in Figure 2, and how the gating network identifies wrong labels in Figure 3).
>
> With these non-trivial improvements, Pi-DUAL outperforms TRAM++ on all 5 benchmarks achieving up to 8.1 accuracy points more than TRAM++ on ImageNet-PI. Pi-DUAL also delivers much better performance in identifying label noise post-training thanks to the explicit modeling of label noise.

---

> ### Author Response · Authors · 2023-11-17
> **Response to Reviewer VkrZ (Part 2)**
>
> **Comparison with results in literature**
>
> We absolutely agree with the reviewer that all methods should be compared under their optimal configurations and this has precisely been our intention in all our experiments. All our experiments, including the reimplementation of other methods, are built on the open-source uncertainty baselines codebase (Nado et al., 2021) and follow as much as possible the benchmarking practices standardized by Ortiz-Jimenez et al. (2023).
>
> As detailed in Appendix B, we follow the same hyper-parameter tuning pipeline for all methods, namely we perform a grid search over their main hyper-parameters using a noisy validation set randomly split from the train set. In this regard, we want to highlight that, since Pi-DUAL only introduces one additional hyper-parameter compared to the cross-entropy baseline, it has the lowest tuning budget compared to the other methods that all have  at least two extra hyper-parameters.
>
> Moreover, we want to clarify that there are no discrepancies between the results we report and those  from the literature, as the reviewer points out. Specifically:
> - On CIFAR10-H, our results for TRAM agree with those released by the original TRAM authors in their follow-up paper (Table 1 in Ortiz-Jimenez et al. 2023). The performance gap between Collier et al. (2022) and Ortiz-Jimenez et al. (2023) is due to the fact that they use two different versions of  CIFAR10-H. Collier et al. (2022) use the “uniform” version while Ortiz-Jimenez et al. (2023) use the harder “worst” version; see discussion in Sec. 2.1 of Ortiz-Jimenez et al. (2023). All our experiments are trained using the latter version of the dataset.
> - In all our tables, to ensure a fair comparison, we report results of the standard ELR and SOP methods which operate without any additional semi-supervised learning techniques. We note that the [leaderboard](www.noisylabels.com) reports separate results for ELR (our baseline) and ELR+ (including semi-supervised learning). For SOP however, we have found that the leaderboard mistakenly reports the results for SOP+ (including semi-supervised learning) instead of SOP, as one can see by comparing to the original SOP paper (cf. Table 4 of Liu et al. (2022)). **Taking this into account, we highlight that our results thus perfectly agree with those reported in the literature, and even exceed them due to better hyper-parameter tuning** (e.g. our ELR baseline is 5 points better on CIFAR100N than the one in the leaderboard).
>
> We hope these clarifications will make the reviewer increase their confidence in our results. We fully understand the initial confusion of the reviewer due to the benchmarking disparities in the literature. Our goal was precisely to ensure a fair comparison of the different methods under the same settings to be able to draw solid conclusions.

---

> ### Author Response · Authors · 2023-11-17
> **Response to Reviewer VkrZ (Part 3)**
>
> **Comparison to semi-supervised methods**
>
> We also agree with the reviewer that a comparison of Pi-DUAL with semi-supervised learning methods would be unfair and that is why we did not consider it in our initial manuscript. Prior works in the noisy-label literature also do not directly compare to DivideMix, but compare DivideMix to versions of their methods which use semi-supervised learning techniques. For example, the authors for ELR compare DivideMix with ELR+, while the authors for SOP compare DivideMix with SOP+ (rather than SOP). ELR+ and SOP+ are respectively more sophisticated versions of the original methods augmented with semi-supervised learning techniques, making the comparison fairer in terms of both costs and complexity.
>
> However, in response to the request of Reviewer h1hP to add a comparison to Divide-Mix and other semi-supervised methods, we have designed an extension of Pi-DUAL, referred to as Pi-DUAL+. It adds to Pi-DUAL two regularization techniques, namely label smoothing and the prediction-consistency regularization from Xie et al., 2020. The latter is a common choice to add semi-supervision to methods of the noisy-label literature (Liu et al., 2020, Cheng et al., 2020). The results for Pi-DUAL+ are shown below and are compared with several prior semi-supervised approaches :
>
> |                |  CIFAR-10H |  CIFAR-10N | CIFAR-100N |
> |----------------|:----------:|:----------:|:----------:|
> |       CE       | 51.10±2.20 | 80.60±0.20 | 60.40±0.50 |
> |   Divide-Mix   | 71.68±0.27 | 92.56±0.42 | 71.13±0.48 |
> |    PES(semi)   | 71.16±1.78 | 92.68±0.22 | 70.36±0.33 |
> |      ELR+      | 54.46±0.50 | 91.09±1.60 | 66.72±0.07 |
> |     CORES*     | 57.80±0.57 | 91.66±0.09 | 55.72±0.42 |
> |      SOP+      | 66.02±0.06 | 93.24±0.21 | 67.81±0.23 |
> | Ours: Pi-DUAL+ | 83.23±0.26 | 93.31±0.21 | 67.99±0.08 |
>
>
> In this table, the baseline results are obtained using the code of the original papers if they are not available in the [public leaderboard](http://www.noisylabels.com). We can see that on CIFAR-10H, Pi-DUAL+ outperforms the rest of the methods by a margin of over 11%, while it performs on par with the state-of-the-art methods on CIFAR-10N and CIFAR-100N. This is because Pi-DUAL+ benefits from high-quality PI on CIFAR-10H, while for CIFAR-10N and CIFAR-100N, the available PI is known to be of poor quality (Ortiz-Jimenez et al., 2023). We have added this table, as well as a detailed description of Pi-DUAL+ in our paper.

---

> > ### Comment · Reviewer_VkrZ · 2023-11-22
> > **Thanks for the response**
> >
> > I thank the authors for their detailed response to address my concerns. I am now confident in the reported accuracy and find the comparison to be fair. Regarding the disparity between Pi-DUAL and TRAM, I concur with the authors that not only does the network structure design differ, but so does the learning objective. While I acknowledge the novelty of Pi-DUAL, I am of the opinion that its technical novelty may not be as substantial as that of TRAM. Therefore, I maintain my score at 6, leaning towards acceptance.

---

### Official Review · Reviewer_sbuM · 2023-11-03

**Soundness:** 3 good
**Presentation:** 3 good
**Contribution:** 3 good
**Rating:** 6
**Confidence:** 3

**Summary:**

This paper introduces the innovative "Pi-DUAL" architecture, which effectively uses privileged information (PI) to distinguish between clean and erroneous labels, offering a crucial solution to label noise. Pi-DUAL demonstrates substantial performance enhancements in several benchmark tests and attains a new state-of-the-art accuracy.  It also excels in identifying noise samples post-training, surpassing other methods.  The ablation study is complete and conducted on all benchmarks for better presentation.

**Strengths:**

First, from the originality point of view, this paper presents a novel architecture, i.e., a noise labeling architecture guided by privileged information (PI), which enables the model to distinguish clean labels and mislabels more clearly. Second, they implement a bidirectional gated output logic structure that decomposes the output logic into a predictive term based on regular input features and a noise-adapted term influenced only by PI. Finally, a PI-driven gating mechanism adaptively chooses between the predictive term and the noise-adaptation term to handle clean and mislabeled learning paths, respectively.
Second, from a significance perspective, the results are impressive, and the improvement of the proposed methods is significant.  Combined with the novelty, I think overall, this is a sound paper introducing an effective method.

**Weaknesses:**

I list several questions that may be helpful.
1. What hardware requirements are needed for Pi-DUAL training for large datasets? If a training cost analysis is provided, I think it can be more useful for deploying your method in more scenarios.
2. How does Pi-DUAL perform in terms of security and privacy protection? For example, is there a risk that privileged information may be compromised? If there is no PI exists, what the performance will be?
3. Are there any parameter tuning for Pi-DUAL for different levels of label noise and datasets? What is your hyper-parameter chosen strategy?

**Questions:**

Please see above weakness.

---

> ### Author Response · Authors · 2023-11-17
> **Response to Reviewer sbuM**
>
> We thank the reviewer for taking the time to review our paper and for all the valuable feedback. We address their comments below:
>
> **Computational cost of Pi-DUAL**
>
> In our experiments, we found that Pi-DUAL does not have extra computational requirements compared for hardware to the standard training baseline. The added cost of processing the PI features (which typically have much fewer dimensions) is indeed negligible with respect to processing the whole images. In this sense, training and inference using Pi-DUAL take roughly the same time as training with standard cross-entropy, TRAM++ or HET. For our large-scale datasets (i.e., ImageNet-PI), we used a TPU V3 with 8 cores for all our experiments.
>
> |                        |   CE   | TRAM++ | Pi-DUAL |   HET  | SOP | ELR |
> |:----------------------:|:------:|:------:|:-------:|:------:|:---:|:---:|
> |  Number of parameters  |   26M  |   32M  |   36M   |   58M  | >1B | >1B |
> | Training time per step | 0.510s | 0.541s |  0.566s | 0.575s |  -  |  -  |
>
> We use the parameter count as a proxy for the memory cost. Since Pi-DUAL models the per-instance level of noise using the PI features, it does not require introducing parameters that scale with the number of samples or classes as SOP and ELR need to. In fact, the extra parameters needed in our experiments to account for the noise and gating networks are comparable to those of TRAM++ and even fewer than the ones introduced by the no-PI method HET.
>
> We have added these comments to the Appendix D of our new version.
>
> **Privacy concerns of using PI**
>
> We thank the reviewer for bringing this topic to our attention and we will make sure to discuss it in our camera-ready work. However, we note that none of our results require the use of personally identifiable annotator IDs. In fact, cryptographically safe IDs in the form of hashes work perfectly fine as PI. In this regard, we do not think that there are serious concerns about possible identity leakages stemming from our work if the proper anonymization protocols are followed. In addition, we have added an ethics statement to the paper regarding the privacy and safety protection requirements of Pi-DUAL.
>
> **Performance without PI**
>
> As mentioned throughout the paper, we would like to emphasize that Pi-DUAL has been explicitly designed to leverage PI features. In this regard, its good performance hinges on the availability of good PI features such as in the case of ImageNet-PI or CIFAR-10H (Ortiz-Jimenez et al. 2023).
>
> However, as mentioned by the reviewer, it is still interesting to study the performance of Pi-DUAL when no PI is available, as we show in Table 4 in the paper (last row). In these cases, we can always construct some randomly-generated PI to train PI-DUAL, following the  procedure of Ortiz-Jimenez et al. (2023). As shown in Table 4 (and copied below for reference), Pi-DUAL with the randomly-generated PI still outperforms the cross-entropy baseline on the CIFAR datasets, but its performance is much worse than when using the real PI of the datasets.
>
> |                        |  CIFAR-10H |  CIFAR-10N | CIFAR-100N | ImageNet-PI (high noise) | ImageNet-PI (low noise) |
> |:----------------------:|:----------:|:----------:|:----------:|:------------------------:|:-----------------------:|
> |           CE           |  51.1±2.2  |  80.6±0.2  |  60.4±0.5  |         68.2±0.2         |         47.2±0.2        |
> |    Pi-DUAL (full PI)   |  71.3±3.3  |  84.9±0.4  |  64.2±0.3  |         71.6±0.1         |         62.1±0.1        |
> | Pi-DUAL (only rand PI) |  53.5±2.2  |  83.7±1.3  |  61.8±0.3  |         68.4±0.1         |         47.0±0.4        |
>
>
> **Hyperparameter tuning strategy**
>
> All our experiments, including the reimplementation of other methods, are built on the open-source uncertainty baselines codebase (Nado et al., 2021) and follow as much as possible the benchmarking practices standardized by Ortiz-Jimenez et al. (2023).
>
> As detailed in Appendix B, we follow the same hyper-parameter tuning pipeline for all methods in which we perform a grid search over their main hyper-parameters using a noisy validation set randomly split from the train set. We do this to make sure that all methods are compared under their optimal and fair settings. In this regard, we want to highlight that, because Pi-DUAL only introduces one additional hyper-parameter being tuned compared to the cross-entropy baseline (i.e., the length of the random PI), it has the lowest tuning budget in comparison to every other method. All other methods introduce at least two additional hyper-parameters to be tuned and therefore we dedicate at least twice as much compute to tune them than the compute used to tune Pi-DUAL. Despite these additional hyper-parameter tuning resources, the performance of Pi-DUAL compares favorably to these methods.
>
> Overall, we see the simplicity of tuning Pi-DUAL compared to other noisy label methods as another strength of the method.

---

### Author Response · Authors · 2023-11-17
**General comment to reviewers**

We kindly thank all the reviewers for their time and for providing valuable feedback on our work, We appreciate that reviewers have pointed out that our results are impressive ([sbuM](https://openreview.net/forum?id=8oYjW8QxuC&noteId=7fcgp9cwrr), [VkrZ](https://openreview.net/forum?id=8oYjW8QxuC&noteId=phW2OkNy6h)) and attest to the efficacy of our method ([h1hP](https://openreview.net/forum?id=8oYjW8QxuC&noteId=s0OISmAZQy)), our method is sound and simple ([sbuM](https://openreview.net/forum?id=8oYjW8QxuC&noteId=7fcgp9cwrr), [VkrZ](https://openreview.net/forum?id=8oYjW8QxuC&noteId=phW2OkNy6h)), our experiments are thorough ([VkrZ](https://openreview.net/forum?id=8oYjW8QxuC&noteId=phW2OkNy6h)) and our paper is clear ([h1hP](https://openreview.net/forum?id=8oYjW8QxuC&noteId=s0OISmAZQy)).

In response to the reviewers we have updated our paper to include a series of new experiments and clarifications to complete the understanding of our method. Specifically:

1. We have performed novel experiments combining Pi-DUAL with unsupervised data augmentation (UDA) and label smoothing to compare its performance to other semi-supervised learning methods from the literature (e.g., DivideMix, ELR+ or SOP+). Our results confirm that even in this more computationally demanding scenario **Pi-DUAL+ outperforms by more than 11.5 accuracy points the state-of-the-art semi-supervised methods when there is good PI available**.
2. We have added extra ablations about the effect of  the model structure of Pi-DUAL, including the model backbone for the prediction network, and the (width, depth) of the noise and gating networks.
3. We have added a new experiment to show that the quality of the PI is the main factor influencing the performance of Pi-DUAL.
4. We have included a new experiment to highlight the differences between Pi-DUAL and TRAM. Specifically, using the TRAM loss with the Pi-DUAL architecture leads to much worse results than using our full method. **This highlights that the innovations of Pi-DUAL do not reside only on the architecture, but also on the principled design of its loss.**
5. We have added a detailed comparison of the runtime and parameter count of different noisy-label methods and showed that Pi-DUAL achieves strong performance improvements at marginal extra compute and memory costs.
6. We have further clarified the settings of our experiments to highlight that **all comparisons are performed using the fairest settings for all methods**. In particular, we even allocate a larger tuning budget to competing methods with more hyperparameters.

We hope that these new results and the clarifications detailed in the individual comments given to each reviewer, will effectively address the concerns raised during the review process. We remain available for engaging in any further discussions that may arise, and we thank you once again for your comments.

---

### Author Response · Authors · 2023-11-22
**Gentle nudge to reviewers**

Dear Reviewers,

Thanks again for the time and effort you have dedicated to reviewing our manuscript. We have provided detailed replies to your comments and we hope our answers and new results have addressed your concerns. Please, let us know if you have any further questions before the discussion period ends.

The authors

---

### Meta-Review · Area_Chair_hM2f · 2023-12-07

**Metareview:**

This paper introduces PI-DUAL, a novel approach designed to address label noise by leveraging privileged information (PI), where the privileged information is only accessible during the training phase. Pi-DUAL demonstrates substantial performance enhancements in several benchmarks and achieve a new state-of-the-art. The ablation study is complete and conducted on all benchmarks for better presentation.

This paper is well written and easy to follow. The reviewers agree that the method presented in this paper is simple and technically sound. However, the reviewers point out that although the improvement of the proposed method is significant in experiments, this paper fails to delve deep into the underlying principles for this improvement. Also, there are some minor issues regarding to the experimental evaluation such as comparing with other baselines and hyperparameter study.

Based on the overall scores, I lean towards a rejection.

**Justification For Why Not Higher Score:**

The reviewers still have some concerns regarding to the novelty and evaluation.

**Justification For Why Not Lower Score:**

Please refer to metareview section.

---

### Decision · Program_Chairs · 2024-01-16

Reject